# Hyperdirect insula-basal-ganglia pathway and adult-like maturity of global brain responses predict inhibitory control in children

Weidong Cai[1]*, Katherine Duberg[1], Aarthi Padmanabhan[1], Rachel Rehert[1], Travis Bradley[1], Victor Carrion[1] & Vinod Menon[1,2,3]*

Inhibitory control is fundamental to children's self-regulation and cognitive development. Here we investigate cortical-basal ganglia pathways underlying inhibitory control in children and their adult-like maturity. We first conduct a comprehensive meta-analysis of extant neurodevelopmental studies of inhibitory control and highlight important gaps in the literature. Second, we examine cortical-basal ganglia activation during inhibitory control in children ages 9–12 and demonstrate the formation of an adult-like inhibitory control network by late childhood. Third, we develop a neural maturation index (NMI), which assesses the similarity of brain activation patterns between children and adults, and demonstrate that higher NMI in children predicts better inhibitory control. Fourth, we show that activity in the subthalamic nucleus and its effective connectivity with the right anterior insula predicts children's inhibitory control. Fifth, we replicate our findings across multiple cohorts. Our findings provide insights into cortical-basal ganglia circuits and global brain organization underlying the development of inhibitory control.

[1] Department of Psychiatry & Behavioral Sciences, Stanford University School of Medicine, Stanford, CA 94305, USA. [2] Department of Neurology & Neurological Sciences, Stanford University School of Medicine, Stanford, CA 94305, USA. [3] Stanford Neuroscience Institute, Stanford University School of Medicine, Stanford, CA 94305, USA. *email: wdcai@stanford.edu; menon@stanford.edu

The ability to adaptively and flexibly control one's actions, including inhibiting prepotent, but inappropriate, responses in favor of goal-directed behavior, is critical for cognitive and affective development. Poor inhibitory action control in childhood is a hallmark of neurodevelopmental disorders including attention deficit hyperactivity disorder and autism[1,2]. Children with better inhibitory control abilities tend to have better academic performance and emotion regulation skills, and are less likely to engage in maladaptive behaviors[3,4]. Studies in adults have shown that inhibitory action control depends critically on the prefrontal cortex and the basal ganglia[5–9]. Here we identify and address critical gaps in our knowledge of brain systems underlying inhibitory action control in children and quantitatively determine whether 'adult-like' functional brain organization predicts superior inhibitory control abilities in children.

Inhibitory action control engages a widely distributed set of cortical and basal ganglia systems[10–16] including, most notably, the right anterior insula (rAI) and the right inferior frontal gyrus (rIFG), and to a lesser extent the right middle frontal gyrus (rMFG), right pre-supplementary motor area (rPreSMA), right supramarginal gyrus (rSMG) and right caudate (rCau)[17]. Both animal[18–21] and human[9,22] studies have also shown strong evidence for the role of the hyperdirect cortical–basal ganglia pathway during inhibitory action control[23,24]. This pathway directly links the subthalamic nucleus (STN) with cortex, which, by bypassing the striatum, confers a significant advantage for rapid stopping of previously initiated actions[25,26]. The STN is a key site of functional convergence of motor circuits[27]: it receives excitatory signals directly from cortical regions and sends excitatory signals to the internal globus pallidus, which in turn inhibits the thalamus (Fig. 1a), thereby resulting in the rapid stopping of a planned action[25]. Although it has been hypothesized that successful stopping requires increased functional engagement of the STN, few studies in adults have found these predicted effects[10,11,14], likely due to inherent difficulties of noninvasively imaging the STN[28]. Consequently, the extent to which the developing brain deploys canonical cortical–basal ganglia systems is not well understood. Furthermore, the extent to which children show adult-like brain responses, and whether the degree of neurofunctional similarities between children and adults predict inhibitory control abilities in children, are also not known. Finally, the role of the STN and the hyperdirect pathway in childhood inhibitory control also remains unknown.

To address these critical questions, we used a classic stop-signal task (SST)[29]. The SST is unique compared to other inhibitory control paradigms because it can be used to compute the stop signal reaction time (SSRT), which measures how fast an individual can stop a prepotent response thereby providing a precise quantitative index of inhibitory control abilities[29]. The SSRT is an optimal measure of stopping as it has a strong association with stopping-related neuronal activity in cortical-STN circuits[30,31]. Behavioral studies have shown rapid developmental change in the SSRT during childhood with more modest changes subsequently between adolescence and adulthood (Fig. 1b and c)[32,33]. Late childhood (aged 9–12) in particular is an important period for the maturation of inhibitory action control abilities[33,34] and impulsive and maladaptive behaviors during this period have increasingly deleterious consequences in adolescence[35,36].

We developed multi-level and multi-scale analytic methods to characterize the neural underpinnings of inhibitory control in late childhood and overcome limitations of previous studies of inhibitory action control in children (Fig. 1d). First, we conducted comprehensive meta-analyses to identify brain regions underlying inhibitory control in middle and late childhood and early adolescence. Second, we acquired event-related functional neuroimaging data from a large cohort of children aged 9–12 years and examined brain activation and connectivity associated with accurate SST performance. Third, we leveraged open-source fMRI datasets from adult participants performing SST to identify reliable and replicable whole-brain activation patterns across two independent datasets. Fourth, we examined correspondence in brain activation patterns between children and adults at multiple levels of spatial organization. Fifth, we developed a neural maturation index (NMI) to determine whether children recruit "adult-like" brain networks during inhibitory control and, critically, to determine whether the extent to which the ability to recruit "adult-like" brain networks is related to each child's inhibitory control ability, using the SSRT. Finally, we replicated key findings using SST data from a large group of children (aged 9–11 years), who participated in the Adolescent Brain and Cognitive Development (ABCD) study[37]. We predicted that greater recruitment of an adult-like inhibitory cortical-basal ganglia control network would be associated with better inhibitory control ability in children, and that SST task-induced activity in the STN and stop-signal-modulated connectivity between STN and right hemisphere cortical regions[17] would predict individual variation in children's inhibitory control abilities. Our findings provide replicable evidence that the hyperdirect cortico-basal ganglia pathway plays a critical role in mediating mature inhibitory control in children.

## Results

**Meta-analysis of fMRI inhibitory control studies in children.** We first conducted a meta-analysis to identify consistent patterns of brain activation associated with inhibitory control in children in order to facilitate direct comparisons with previous meta-analyses of adult functional neuroimaging studies of inhibitory control[17]. We identified published fMRI studies of the SST that reported whole-brain activation coordinates in typically developing children spanning middle and late childhood and early adolescence (6–13 years old). Due to the paucity of studies using SST paradigms in the neurodevelopmental brain imaging literature, we also included studies using the Go/NoGo task (GNGT), another canonical, but less precise, inhibitory control task. As of November 2017, our search resulted in 14 studies, with 2 that used the SST and 12 that used the GNGT, that meet the inclusion criteria (see the "Methods" section, Supplementary Tables 1 and 2).

Adults showed a highly consistent pattern of brain activations associated with inhibitory control[17], encompassing bilateral AI, right IFG, MFG, preSMA, posterior parietal cortex (PPC), and striatum ($p < 0.01$, FWE corrected, Fig. 2a, also see cluster coordinates in Supplementary Table 3). Children showed a similar pattern but at a lenient threshold ($p < 0.01$, uncorrected) that overlapped with previous meta-analytic results from studies in adults[17], including AI, IFG, MFG, preSMA, and PPC (Fig. 2b, also see cluster coordinates in Supplementary Table 4).

These results suggest that, in contrast to adults, extant functional neuroimaging studies of inhibitory control in children have been much less consistent. Possible factors include small number of studies that meet inclusion criterion, small sample sizes, more variable activation patterns in children, wide age ranges of participants across studies, and limited number of studies using the SST.

**Inhibitory action control in children during SST performance.** To overcome weaknesses in the extant literature identified by the above meta-analysis, we analyzed three datasets from 38 children and 42 adults who performed the SST during fMRI scanning. The first dataset (Stanford_Child) was acquired at Stanford University

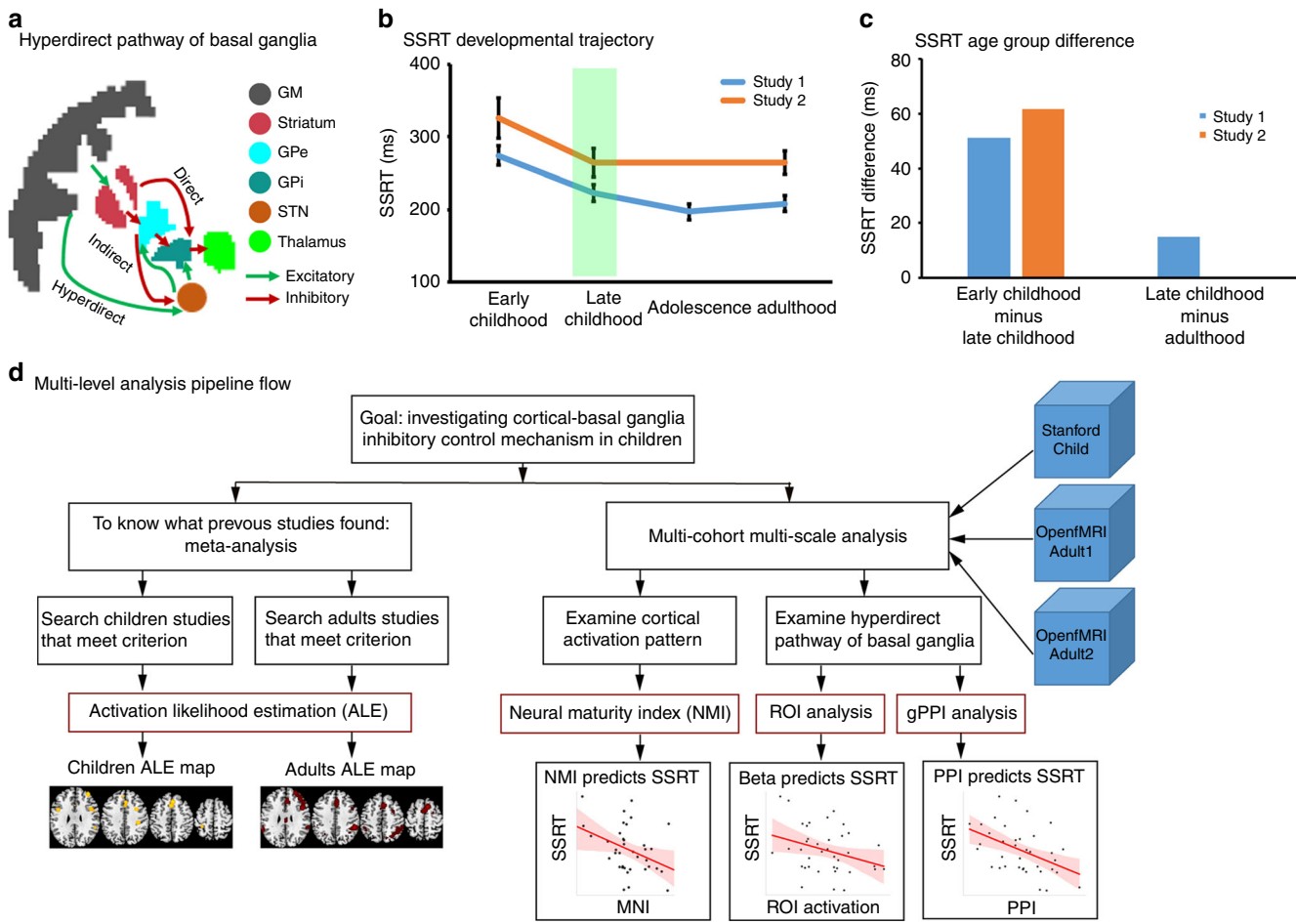

**Fig. 1** Behavior, cortico-basal ganglia pathways and pipeline. **a** Illustration of three cortico-basal ganglia pathways, including the hyperdirect pathway examined in the present study. The hyperdirect pathway is hypothesized to play a crucial role in rapid stopping. **b** Behavioral improvements in inhibitory action control ability measured using stop signal reaction time (SSRT) during childhood revealed by previous developmental studies. **c** Changes in stop signal reaction time (SSRT) from early to late childhood are much larger than between late childhood and adulthood. Panels (**a**) and (**b**) were generated using data from study 1 published by Schachar et al. [32] and study 2 by Williams et al. [33]. We grouped data from grade 4 and grade 6 in study 1 to match the late childhood group in study 2. **d** Overview of data analysis pipeline used in the current study. We first conducted meta-analyses to determine whether previous studies in children have yielded consistent findings as studies in adults. We then used three event-related fMRI datasets (Stanford_Child, OpenfMRI_Adult1, and OpenfMRI_Adult2) to investigate adult-like maturity of inhibitory action control during the stop signal task (SST). Our analyses focused on similarity of brain activation patterns between children and adults during the SST. We developed a neural maturity index (NMI) to quantify brain activation similarity between children and adults. We also investigated neural signatures of the hyperdirect pathway in children by examining the relationship between task-induced activation in the subthalamic nucleus (STN) and SSRT and task-related cortical effective connectivity with the STN and SSRT

from 38 children, aged 9–12 years (see neuropsychological scores in Supplementary Table 5). Each child completed two runs of a standard event-related SST. The second and third fMRI datasets were based on two independent studies in adults obtained from OpenfMRI (openfMRI.org; OpenfMRI_Adult1: $N = 18$; OpenfMRI_Adult2: $N = 24$). Each adult completed two runs of a standard event-related SST.

SSRT was used to assess children's inhibitory control function (Table 1). We confirmed that behavioral data of each child included in the analysis did not violate the Race Model[29,38]. For example, a key requirement is that Unsuccessful Stop trials have shorter reaction times than Go trials. The SSRT in the Stanford_Child cohort was $299 \pm 53$ ms (mean ± standard deviation), consistent with previous behavioral studies (Fig. 1b, and c)[32,33]. SSRTs in children were longer by over 100 ms compared to both adult cohorts: $193 \pm 51$ ms (OpenfMRI_Adult1) and $167 \pm 67$ ms (OpenfMRI_Adult2). Due to differences in task settings, SSRTs were not compared directly between groups.

**Brain activation during inhibitory control in children**. We investigated event-related fMRI activation associated with the SST in children using a general linear model. Events of interest included go trials (Go), successful stop trials (SuccStop), and unsuccessful stop trials (UnsuccStop). We contrasted brain responses to Succ-Stop versus Go to determine brain activations associated with inhibitory control[11,39]. Children showed significantly greater activation in bilateral AI, right IFG, right MFG, preSMA, and PPC ($p < 0.01$, FDR corrected, Fig. 3a, Supplementary Table 6). Moreover, these regions showed close correspondence with ALE patterns from our meta-analysis of studies in adults (Fig. 2a).

The OpenfMRI_Adult1 and OpenfMRI_Adult2 datasets were also analyzed using the same processing steps as the Stanford_Child dataset. SuccStop versus Go trials revealed significantly greater activation in bilateral AI, right IFG, MFG, preSMA, and PPC ($p < 0.01$, FDR corrected) in both the openfMRI_Adult1 (Fig. 3b, Supplementary Table 7) and openfMRI_Adult2 cohorts (Fig. 3c, Supplementary Table 8).

**a**

Meta-analytic results in adults (*p* < 0.01, FWE corrected)

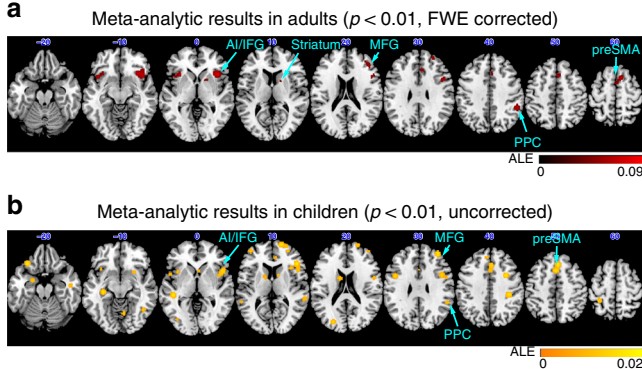

**b**

Meta-analytic results in children (*p* < 0.01, uncorrected)

**Fig. 2** Meta-analysis of neuroimaging studies in adults and children. **a** ALE Meta-analysis of 70 inhibitory control neuroimaging studies in adults (34 SST, 36 GNGT) revealed consistent activation of bilateral anterior insula (AI), right inferior frontal gyrus (IFG), middle frontal gyrus (MFG), pre-supplementary motor area (preSMA), posterior parietal cortex (PPC) and striatum (*p* < 0.01, FWE corrected). **b** ALE Meta-analysis in 14 studies in children (2 SST, 12 Go/NoGo) revealed activation of AI, IFC, MFG, preSMA and PPC but only at a weaker threshold (*p* < 0.01, uncorrected). Weak meta-analytic results in children reflect inconsistent and/or weak findings inform previous neuroimaging studies of inhibitory control in children. SST stop-signal task; GNGT Go/NoGo task

### Table 1 Summary of behavioral performance

|  | Mean ± std |
| --- | --- |
| Go Accuracy (%) | 94 ± 4 |
| Go RT (ms) | 504 ± 78 |
| Stop Accuracy (%) | 51 ± 6 |
| Stop Fail RT (ms) | 436 ± 52 |
| SSD (ms) | 187 ± 69 |
| SSRT (ms) | 299 ± 53 |

**Children show adult-like global brain activity**. To determine whether children and adults engage overlapping brain-wide activation patterns associated with inhibitory control, we used a template including all gray matter voxels except sensorimotor cortex. The sensorimotor cortex was excluded because the neural response pattern from these voxels are largely dependent on the specific choice of sensory stimuli and motor response in a given study, which may over-estimate or under-estimate the similarity of activation patterns between groups. We found a strong positive correlation between brain activation patterns elicited during inhibitory control (SuccStop versus Go contrast) in the Stanford_Child and OpenfMRI_Adult1 cohorts ($r = 0.69$, $p < 0.001$, permutation test), and in the Stanford_Child and OpenfMRI_Adult2 cohorts ($r = 0.75$, $p < 0.001$, permutation test) (Fig. 3d). These relations were weaker in contrasts involving Go and UnsuccStop trials (Supplementary Note, Supplementary Figs. 1 and 2). These results suggest that children and adults show similar global activation patterns associated with inhibitory control.

**Children show adult-like cognitive control network activity**. Using the parcellation scheme of Power et al.[40], we then determined whether children and adults engage overlapping activation patterns within cognitive control networks known to be involved in a wide range of cognitive tasks (Fig. 4a). Again, to minimize the impact of sensory stimuli and motor output, Regions of Interest (ROIs) from sensorimotor networks were excluded. We found a

strong positive correlation between brain activation patterns in core cognitive control networks Stanford_Child and Open-fMRI_Adult1 cohorts ($r = 0.76$, $p < 0.01$, permutation test) and between Stanford_Child and OpenfMRI_Adult2 cohorts ($r = 0.80$, $p < 0.01$, permutation test) (Fig. 4b) across these ROIs. Moreover, activation patterns revealed that the salience network (SN), the frontoparietal network (FPN), and the ventral attentional network (VAN) showed the greatest activation during stopping in children and adults, whereas the default mode network (DMN) showed the greatest deactivation or the least activation in children and adults (Fig. 4c). These results suggest that children and adults show similar activation patterns associated with inhibitory in multiple cognitive control networks.

**Children show adult-like inhibitory control system activity**. Next, we focused our analysis on core cortical-basal ganglia inhibitory control regions previously identified in adults during inhibitory control tasks[10–16]. Regions of Interest (ROIs) were defined by a previous meta-analysis of neuroimaging studies of inhibitory control in adults[17], including rAI, rIFG, rPreSMA, rMFG, rSMG, and rCau (Fig. 5a, Supplementary Table 9). We also included ROIs in the rSTN and lSTN, based on a previous ultra-high-resolution 7T MRI study[28], which provided more accurate localization of this small subcortical structure.

We examined whether these independently defined ROIs have significantly greater activation during SuccStop than Go trials in children. Paired *t*-tests revealed significantly greater activation during SuccStop relative to Go trials in rAI, rIFG, rPreSMA, and rSMG (all *p*'s < 0.05, Bonferroni corrected) (Fig. 5b). The rCau and rMFG had greater activation in SuccStop relative to Go trials, but these differences did not survive correction for multiple comparisons ($p < 0.05$, uncorrected). The difference between SuccStop and Go was not significant in bilateral STN ($p > 0.05$). Figure 5b also shows comparable findings from the two cohorts of adults.

Finally, we examined whether children and adults have similar patterns of activation in these regions during stopping. We examined the relationship in ROI activation patterns between children and adult cohorts during stopping (SuccStop vs. Go) and found a significant correlation between Stanford_Child and OpenfMRI_Adult1 cohorts ($r = 0.90$, $p < 0.01$, permutation test) and between Stanford_Child and OpenfMRI_Adult2 cohorts ($r = 0.94$, $p < 0.01$, permutation test) (Fig. 5c).

Taken together, these results provide quantitative evidence that brain activation patterns associated with inhibitory control in children are similar to those in adults across multiple levels of analysis and topographical granularity, including those that are central to the SST.

**Neural maturity index (NMI) predicts children's inhibitory control abilities**. We next tested the hypothesis that the more adult-like, or mature, brain activation a child has, the better the child's inhibitory control abilities. We used an NMI which measures the similarity of brain activation patterns between children and adults. The NMI was defined as the spatial correlation between an individual child's brain activation map and an adult group-level brain activation map (adult reference map) (Fig. 6a). A higher NMI indicates a greater similarity in brain-wide activation patterns between children and adults.

We found a significant negative correlation between the NMI and SSRT using a reference map derived from the Open-fMRI_Adult1 reference map ($r = -0.32$, $p = 0.05$, Cohen's $d = 0.67$, Pearson's correlation) (Fig. 6b). We replicated this finding using the OpenfMRI_Adult2 reference map ($r = -0.39$, $p < 0.05$, Cohen's $d = 0.85$, Pearson's correlation) (Fig. 6b). To

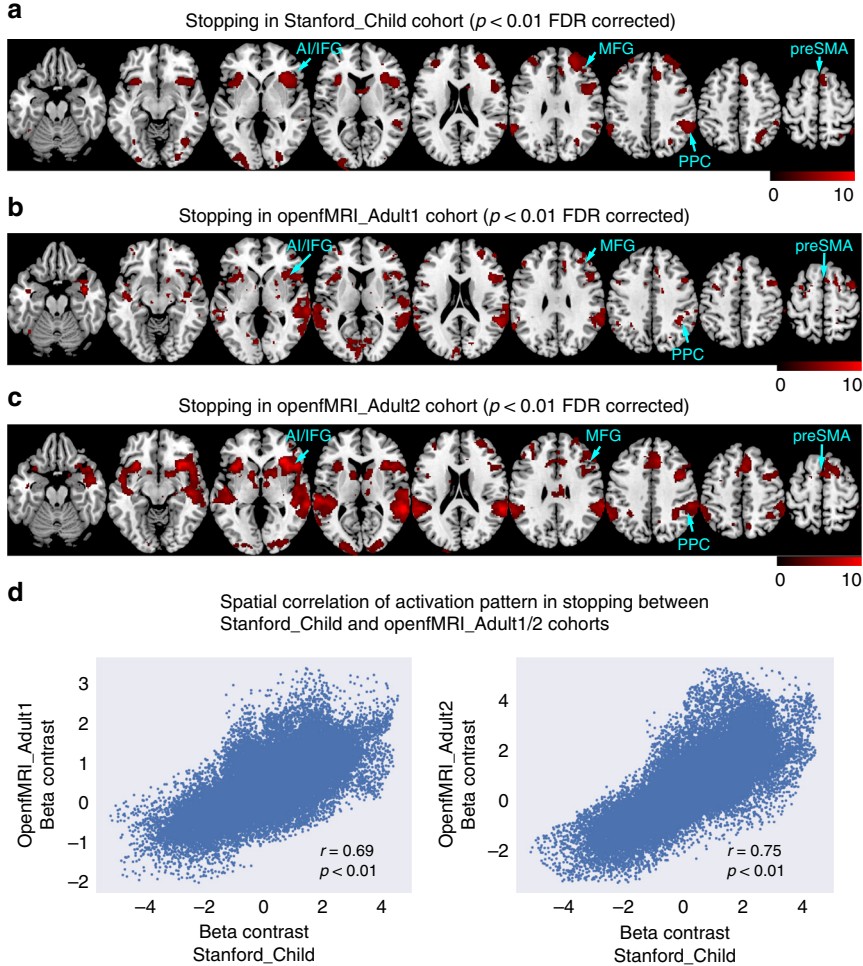

**Fig. 3** Similarity of activation patterns in children and adults. **a** Stopping-related (SuccStop versus Go) activation of bilateral anterior insular (AI), right inferior frontal gyrus (IFG), middle frontal gyrus (MFG), pre-supplementary motor area (preSMA), and posterior parietal cortex (PPC) in children from the Stanford_Child cohort ($p < 0.01$, FDR corrected). **b**, **c** The two adult cohorts (OpenfMRI_Adult1 and OpenfMRI_adult2) showed activation in similar brain regions including bilateral AI, right IFG, MFG, preSMA, and PPC in stopping. **d** Brain-wide activation patterns elicited by stopping in children were significantly correlated with activation patterns in two different adult cohorts. Source data are provided as a Source Data file

further examine whether this relationship was driven by other potential confounds, we conducted multiple linear regression with SSRT as the dependent variable and NMI, age, gender, and maximum head motion displacement as independent variables. We found that the NMI was the most robust predictor in both the OpenfMRI_Adult1 ($p = 0.07$) and OpenfMRI_Adult2 ($p < 0.05$) cohorts (Table 2). Results provide replicable evidence that children with greater adult-like brain activation patterns demonstrate more efficient inhibitory control.

To investigate the developmental specificity of our findings, we tested whether adults closer to the average adult template also show faster SSRTs. However, we did not find a significant effect in either adult cohort (OpenfMRI_Adult1: $p = 0.17$; OpenfMRI_Adult2: $p = 0.20$, Pearson's correlation), highlighting the developmental specificity of the NMI.

**Children show dissociable activation patterns in the rSTN**. To detect differential STN responses associated with inhibitory control, we examined both (univariate) activation levels and multivariate patterns elicited by SuccStop and Go stimuli. Children did not show significant differences in STN activation levels elicited by SuccStop and Go stimuli. To decode multivariate patterns associated with inhibitory control, we trained a linear

support vector machine model and determined cross-validation accuracy for distinguishing between multivoxel responses elicited by SuccStop and Go stimuli. In the Stanford_Child cohort, cross-validation accuracies were significant in the right STN 59% (accuracy = 59%, $p = 0.05$, permutation test) but not the left STN ($p = 0.26$, permutation test). These results suggest that the right STN shows distinct neural responses to SuccStop and Go trials in children.

Results from similar analyses in the two adult cohorts are described in the Supplementary Notes.

**STN activation predicts inhibitory control in children**. To investigate the involvement of the hyperdirect basal ganglia pathway in inhibitory control we examined the relation between STN activation during inhibitory control and SSRT[10,11,14] in the Stanford_Child cohort. Because preferential STN activation in SuccStop and UnsuccStop remains controversial[23,41], here we averaged STN activation during SuccStop and UnsuccStop trials.

We found a negative correlation between STN activation and SSRT in the Stanford_Child cohort (rSTN: $r = -0.32$, $p = 0.05$, Cohen's $d = 0.67$; lSTN: $r = -0.32$, $p = 0.05$, Cohen's $d = 0.67$, Pearson's correlation) (Fig. 7a, Supplementary Table 10). To further examine whether this relationship was driven by potential

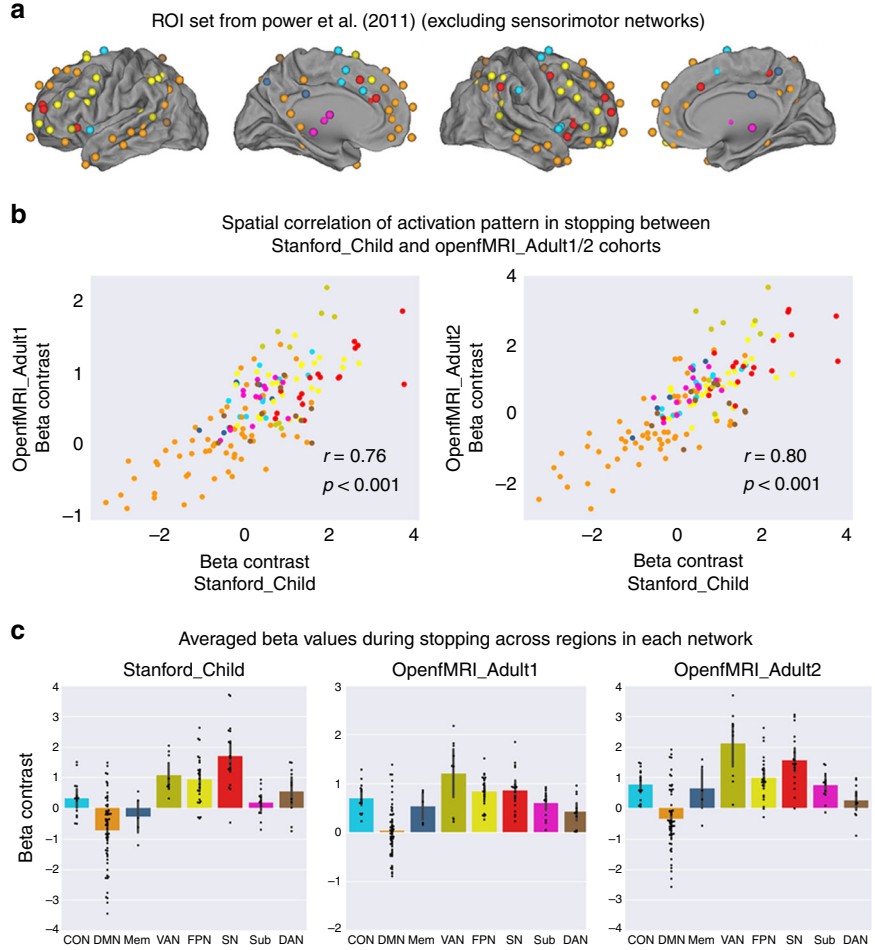

**Fig. 4** Similarity of network activation in children and adults. **a** ROIs were defined using functional networks implicated in cognition, perception, and control from an independent study by Power et al.[40]. Each ROI was color-coded based on the functional network to which it belongs. ROIs from sensorimotor networks were excluded in the analysis. **b** Brain activation patterns elicited by stopping (SuccStop versus Go) in children were significantly correlated with activation patterns in the two different adult cohorts. SN, Sub, FPN and VAN nodes showed the greatest activation during stopping in children and adults. DMN nodes showed the greatest deactivation during stopping in children and adults. **c** Activation levels (Beta-weights) in large-scale brain networks during stopping in children and adults. Error bars represent standard deviation. Activations were averaged across regions in each functional networks. Networks are color-coded and the same coding was used across all the panels: CON: cyan; DMN: orange; Mem: navy blue; VAN: green yellow; FPN: yellow; SN: red; Sub: magenta; DAN: brown. CON: cinguo-opercular network, DMN: default mode network, Mem: memory network, VAN: ventral attention network, FPN: fronto-parietal network, SN: salience network, Sub: subcortical network, DAN, dorsal attention network. Error bar stands for standard deviation. Source data are provided as a Source Data file

confounds, we conducted multiple linear regression with SSRT as the dependent variable and STN activation, age, gender, and maximum head motion displacement as independent variables. We found that rSTN activation was the only significant predictor of SSRT ($p = 0.03$, general linear regression) after controlling effects of age, gender, and head motion (Table 3). These results suggest a key role of the STN in modulating inhibitory control in the developing brain.

Results from similar analyses in the two adult cohorts are described in the Supplementary Tables 11 and 12.

**Cortical-STN connectivity predicts inhibitory control in children.** We next investigated the role of hyperdirect cortical-STN circuits in inhibitory control[10,17], focusing on right hemisphere cortical regions rAI, rIFG, rPreSMA identified above in the meta-analysis[17] (Fig. 2a). We examine whether task-modulated functional connectivity between these cortical regions and the rSTN predicted SSRTs in children. We used general psychophysiological interaction (gPPI) to measure connectivity changes between cortical seed regions (rAI, rIFG, rPreSMA) and rSTN in SuccStop versus UnsuccStop trials. We chose this contrast because the most notable cortical regions in the inhibitory control network, i.e. rAI and rIFG, have dissociable activation profiles in SuccStop and UnsuccStop[17], whereas preferential STN activation in SuccStop and UnsuccStop remains controversial[23,41]. Therefore, we specifically focused on cortical-STN interaction in SuccStop versus UnsuccStop.

We found that stop signal-related connectivity between the rAI and rSTN, but not the lSTN, was significantly correlated with SSRT ($r = -0.46$, $p < 0.005$, Cohen's $d = 1.04$, Pearson's correlation) in children, such that increased rAI–rSTN connectivity was associated with faster SSRTs (Fig. 7b). Connectivity between the other prefrontal cortical regions and STN were not significantly correlated with SSRT ($p > 0.05$, Pearson's correlation). Multiple linear regression analyses, which included SSRT as the dependent variable and STN activation, age, gender, and maximum head motion displacement as independent variables, confirmed that task-modulated connectivity between the rAI and rSTN was the best predictor of SSRT ($p < 0.01$, general linear regression) (Table 4, also Supplementary Table 13).

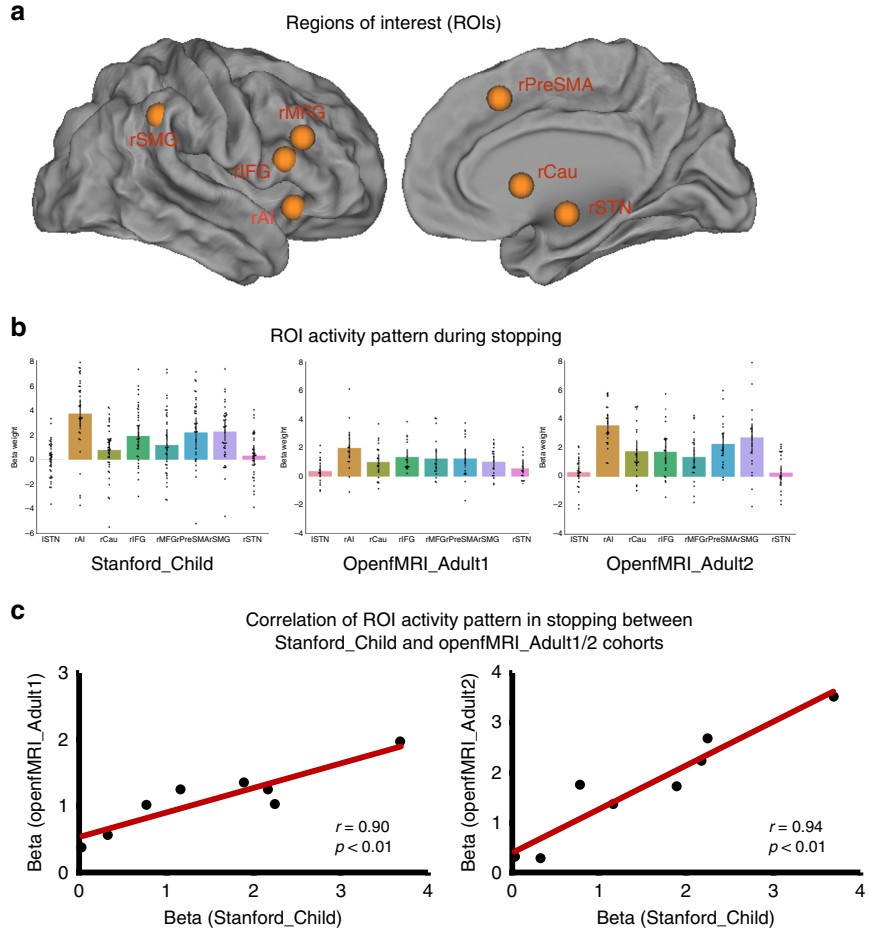

**Fig. 5** Similarity of regional activation in children and adults. **a** Regions of interest (ROIs), as determined by a previous meta-analysis of the stop signal task (SST) in adults[17]. ROIs consisted of right anterior insula (rAI), right inferior frontal gyrus (rIFG), right middle frontal gyrus (rMFG), right pre-supplementary motor area (rPreSMA), right supramarginal gyrus (rSMG), and right caudate (rCau). Left and right subthalamic nuclei (lSTN and rSTN) ROIs were based on a high-resolution 7T structural MRI study[28]. **b** ROI analyses revealed greater activation in stopping (SuccStop versus Go) in rAI, rCau, rIFG, rMFG, rPreSMA and rSMG in children and two adult cohorts (Bonferroni-corrected for multiple comparisons). Error bars represent standard deviations. **c** ROI activation pattern elicited by stopping in children was significantly correlated with activation patterns in two different cohorts of adult. Source data are provided as a Source Data file

These results highlight a specific role of insula-STN circuits in implementing stopping actions in children and reveal a source of individual differences in their inhibitory control abilities.

Results from similar analyses in the two adult cohorts are described in the Supplementary Note.

**Robustness with respect to head motion**. To examine the robustness of our findings with respect to head motion, we conducted additional analyses using a volume repair method (Supplementary Methods) and replicated all of the key findings (Supplementary Note, Supplementary Figs. 3–5, Supplementary Tables 14–16).

**Robustness with respect to age range in adult cohorts**. To examine whether our findings are impacted by the wider age range in the adult dataset, we conducted additional analyses using only adults within a restricted age range (18–26 years) and replicated all of the key findings (Supplementary Methods, Supplementary Note, Supplementary Fig. 6, Supplementary Table 17).

**Robustness with respect to neuropsychological regressors**. To assess whether other potential confounds can account for individual difference in SSRT in children, we conducted additional

analyses by adding neuropsychological measures in multiple linear regression analyses and replicated all of the key findings (Supplementary Methods, Supplementary Note, Supplementary Tables 18–20).

**Replication using SST fMRI data from the ABCD study**. To further demonstrate the robustness of our findings, we analyzed SST fMRI and behavioral data from 186 children aged 9–11 in the NIH-funded Adolescent Brain and Cognitive Development (ABCD) study[37]. Here, we summarize the main replication findings. Details of the dataset and methods are in Supplementary Methods.

First, we examined the relationship between NMI and SSRT in the ABCD dataset. We found a significant negative correlation between the NMI and SSRT using the OpenfMRI_Adult1 reference map ($r = -0.15$, $p < 0.05$, Cohen's $d = 0.30$, Pearson's correlation) (Supplementary Fig. 7). To further examine whether this relationship was driven by other potential confounds, we conducted multiple linear regression with SSRT as the dependent variable and NMI, age, gender, and maximum head motion displacement as independent variables. We found that NMI was the most robust predictor of SSRT ($p = 0.04$) (Supplementary Table 21).

**a** Neural maturity index (NMI)

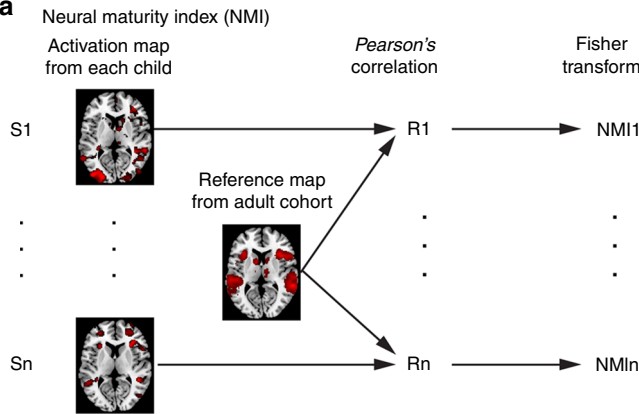

**b** NMI is correlated with stopping ability in children

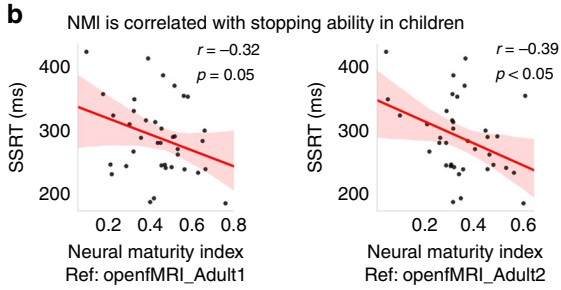

Fig. 6 NMI predicts stopping action in children. **a** Schematic illustration of the algorithm used to compute a neural maturity index (NMI) in each child. We computed the spatial correlation between each child's activation map and a reference activation map from each of the adult cohorts using Pearson's correlation and a Fisher transformation was used to determine the NMI for each child. **b** NMI in children was negatively correlated with stop signal reaction time (SSRT). This relationship was replicated using reference maps from the two adult cohorts. Source data are provided as a Source Data file

**Table 2 Linear regression shows that NMI predicts children's SSRT**

|  | beta | t | p |
|---|---|---|---|
| *Reference: OpenfMRI_Adult1* |  |  |  |
| Neural maturity index | −108.9 | −1.86 | 0.07 |
| Age | −10.3 | −0.58 | 0.56 |
| Gender | −6.9 | −0.35 | 0.73 |
| Maximum frame-wise displacement | −2.8 | −0.25 | 0.8 |
| *Reference: OpenfMRI_Adult2* |  |  |  |
| Neural maturity index | −163.2 | −2.37 | 0.02* |
| Age | −14.2 | −0.83 | 0.42 |
| Gender | −2 | −0.1 | 0.92 |
| Maximum frame-wise displacement | −4.7 | −0.43 | 0.67 |

*$p < 0.05$

Second, we examined the relationship between STN activation during stopping and SSRT in the ABCD dataset. We found a negative correlation between STN activation in stopping and SSRT (rSTN: $r = −0.20$, $p = 0.005$, Cohen's $d = 0.41$, lSTN: $r = −0.22$, $p = 0.005$, Cohen's $d = 0.45$; Pearson's correlation) (Supplementary Fig. 8). Multiple linear regression analyses with SSRT as the dependent variable and STN activation, age, gender, and maximum head motion displacement as independent variables, confirmed that STN activation was the most robust predictor of SSRT (rSTN: $p = 0.0006$, lSTN: $p = 0.0002$; general linear regression) (Supplementary Table 22).

**a** Task-induced activity in stopping is correlated with SSRT

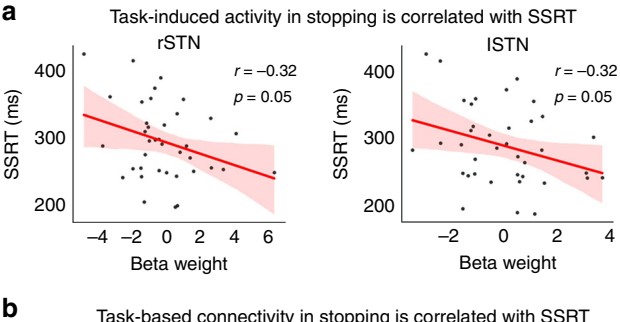

**b** Task-based connectivity in stopping is correlated with SSRT

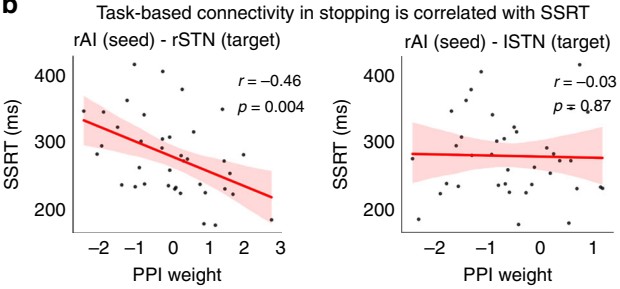

Fig. 7 STN activity and connectivity predict stopping in children. **a** Activation in the STN during stopping was negatively correlated with SSRT in children. **b** Effective connectivity between the rAI (seed) and rSTN (target) was negatively correlated with SSRT in children. No such relation was observed in the left hemisphere. Source data are provided as a Source Data file

**Table 3 Linear regression shows that rSTN activation predicts children's SSRT**

|  | beta | t | p |
|---|---|---|---|
| Activation in rSTN | −4.4 | −2.05 | 0.04* |
| Age | −14.1 | −0.8 | 0.42 |
| Gender | −5.4 | −0.28 | 0.78 |
| Max frame-wise displacement | −0.3 | −0.03 | 0.98 |
| Activation in lSTN | −4.8 | −1.86 | 0.07 |
| Age | −9.1 | −0.51 | 0.62 |
| Gender | −1.8 | −0.1 | 0.93 |
| Max frame-wise displacement | −1.3 | −0.12 | 0.91 |

*$p < 0.05$

**Table 4 Linear regression shows that rAI–rSTN connectivity predicts children's SSRT**

|  | beta | t | p |
|---|---|---|---|
| Effective connectivity between rAI and rSTN during stopping | −20.3 | −2.93 | 0.006** |
| Age | −11.3 | −0.68 | 0.5 |
| Gender | 2.4 | 0.13 | 0.9 |
| Maximum frame-wise displacement | 2.4 | 10.33 | 0.82 |
| Effective connectivity between rAI and lSTN during stopping | −3 | −0.31 | 0.76 |
| Age | −12.3 | −0.65 | 0.52 |
| Gender | −4.7 | −0.65 | 0.82 |
| Maximum frame-wise displacement | 0.5 | 0.04 | 0.97 |

**$p < 0.01$

Third, we examined the relationship between rAI–rSTN effective connectivity and SSRT in the ABCD dataset. We found that stop signal-modulated effective connectivity between the rAI and rSTN, but not the lSTN, during stopping was significantly

correlated with SSRT ($r = -0.16$, $p < 0.05$, Cohen's $d = 0.32$, Pearson's correlation) in children (Supplementary Fig. 9), such that increased rAI–rSTN connectivity was associated with faster SSRTs. Multiple linear regression analyses, which included SSRT as the dependent variable and effective connectivity, age, gender, and maximum head motion displacement as independent variables, confirmed that task-modulated connectivity between the rAI and rSTN was the best predictor for SSRT ($p = 0.03$, general linear regression) (Supplementary Table 23).

In sum, analysis of task fMRI and behavioral data from children aged 9 to 11 in the ABCD study replicated all major results from the Stanford_Child cohort, demonstrating the robustness of our findings.

## Discussion

Late childhood, spanning pre-adolescence ages 9–12, is important for the maturation of inhibitory control[33]. We used a multi-level analytic approach with three independent datasets (Fig. 1d) to investigate mature adult-like inhibitory control brain mechanisms and their relation to individual differences in children's inhibitory control abilities and importantly replicated our main findings using data from the ABCD study[37]. The use of open-source fMRI datasets allowed us to test the generalizability of our findings, and to advance analytic approaches for facilitating replicability in developmental cognitive neuroscience. We provide replicable evidence that the hyperdirect insula–basal ganglia pathway plays a key role in the development of mature inhibitory control, and that adult-like maturity of brain responses predicts individual differences in inhibitory control in children.

Our meta-analysis identified several gaps in the extant literature on the neural basis of inhibitory control in children, including identification of just 14 fMRI studies of inhibitory control compared to 70 studies identified in adults, small sample sizes (Supplementary Table 1), and choice of tasks that preclude the ability to compute stopping times, such as the GNGT. As a result, there has been poor convergence between studies of inhibitory control involving SST in adults on the one hand, and those based on GNGT in children on the other hand. Our analytic strategy and use of SST fMRI data from two open-source adult cohorts and two large groups of child cohorts attempted to overcome these limitations.

Our child cohort, which spanned late childhood, as a group showed successful performance on the SST and accurate sustained attention (high Go Accuracy, Table 1). However, their SSRTs ($299 \pm 53$ ms) were longer compared to adults' SSRTs in similar experimental settings (under 200 ms)[33]. SSRTs in children were 100 ms and two standard deviations longer than those reported in similar fMRI studies of adult participants[11–13,42]. The distribution of behavioral and cognitive-affective abilities in our child cohort were in the normative range of neuropsychological function as assessed using multiple measures of IQ, achievement, and cognitive, and emotional control capacity (Supplementary Table 5). These results suggest that in a sample that is representative of typically developing children ages 9–12 years, inhibitory control mechanisms are not fully mature by late childhood.

To determine the extent to which children have developed adult-like inhibitory control systems, we first identified reliable patterns of brain activation in regions associated with inhibitory control[17] in two independent adult cohorts. These results confirm that increased activity in key nodes of the inhibitory control network is a replicable finding in children and adults.

Our multilevel analytic approach provided strong evidence that children can recruit key nodes of adult-like cognitive and inhibitory control networks during performance of inhibitory control task. Specifically, we demonstrated significant and strong relationships in task-related engagement between children and adults first using whole-brain activation patterns, second, using an independent set of brain-wide ROIs involved in task-general cognitive control networks[40], including activation in regions of the salience, subcortical, frontal–parietal, and ventral attention networks (VANs), and deactivation of the DMN during inhibitory control, and third, using independently defined ROIs from task-specific cortical–basal ganglia inhibitory control regions that included the rAI, rIFG, rMFG, rPreSMA, rSMG, rCau, and bilateral STN ROIs. Taken together, these results provide and robust evidence for a high level of correspondence in brain activation between children and adults spanning multiple levels of analysis, topographical granularity and organization, and functional specificity.

NMI, which assesses similarity of inhibitory-control-related brain activation patterns between children and an adult template, provided a quantitative metric for assessing whether adult-like brain responses can predict SSRT in children. We found a significant negative correlation between NMI and SSRT, suggesting that the more similar a child's inhibitory control network is to adults, the better their ability to stop a prepotent response. These results demonstrate that brain activation patterns from adult participants provide robust and replicable templates for quantifying individual differences in the maturation of brain systems involved in inhibitory control. Critically, we demonstrate that individual variations in the level of mature brain activity are an important source of heterogeneity in children's inhibitory control ability. There are two possible explanations for our findings: (1) developmental changes drive the effect that children with better performance show more "adult-like" brain activity patterns; or (2) that any participant with a more "typical" brain activity pattern related to stopping would show a better SSRT, such that our NMI results are not related to development or maturity. To address this question, we conducted a similar analysis within the adult samples and found that adults did not show a relation between NMI and SSRT. These results support the first hypothesis, and suggest that NMI results are related to development and maturation. Studies using NMI measures, as developed here, and longitudinal data are needed to further disentangle sources of individual variability over development. Our findings provide an ecologically valid template for new research directions in this regard.

The next main goal of our study was to specifically probe the hyperdirect insula–basal ganglia pathway in children, focusing on the STN plays an important role in countermanding planned actions[18–21,26]. We found that although STN activation levels did not differ between Successful Stop and Go trials, children showed dissociable activation patterns in the rSTN such that multivariate patterns of activation were able to discriminate between these task conditions. Crucially, we also found that the more the rSTN is activated during inhibitory control, the faster a child can stop a prepotent response, highlighting a role for the STN in efficient action control during childhood.

The hyperdirect pathway connects cortical areas directly to the STN, bypassing the striatum, thereby facilitating faster stopping behaviors[25]. Consistent with this model we found that the strength of SST-modulated connectivity between the rAI and rSTN was correlated with inhibitory control abilities in children. This result held after FDR-correction for multiple comparison, which included two other prefrontal cortex regions that have also been implicated in stopping and have extensive connections with the STN[10,28,43,44].: rIFG, and rPreSMA, highlighting the specificity of the STN-AI connection during stopping. Structural imaging studies in adults have revealed that the strength of white matter pathways between prefrontal regions, including IFG and preSMA, and STN during inhibitory control is related to

SSRT[24,28]. Crucially, the STN and AI have direct anatomical connections[43,45], but their functional interactions have not been previously examined. Such a role for the AI is not surprising given that it is the most strongly and consistently activated brain region during inhibitory control in adults[17] and is important for detection of salient cues that signal the need for inhibitory control[17,46,47]. We also found that the right AI was the most strongly activated among all nodes of the inhibitory control network in both adults and children (Fig. 5b). Taken together, our findings suggest that rAI-based detection of salient Stop cues and Stop-signal-specific interactions of the rAI with the rSTN may facilitate fast stopping action in children.

Resolving the precise role of the STN and its functional circuits in stopping action control in humans using BOLD-fMRI signals has been challenging because of its small size. A particular complexity with regards to parsing STN engagement during the SST is that its response is sensitive to the urgency to stop a prepotent response. Thus, for example, recent studies have paradoxically revealed greater STN activation in Unsuccessful relative to Successful stopping[14,23,41]. Furthermore, the STN is functionally heterogeneous with only its ventromedial subdivision exhibiting neuronal activity specifically aligned with inhibitory action control[20]. Indeed previous fMRI studies in adults have reported mixed or weak findings with regards to STN activity during stopping[10,11,14,23,41]. We suggest that multivariate activation patterns in the rSTN might be better suited for distinguishing responses to different trial types. In any case, our findings demonstrate a clear role for the rSTN in the developing brain.

Previous neuroimaging studies have shown that inhibitory action control relies on multiple prefrontal and basal ganglia regions[10–12,14,15,39,42,48]. In particular, the rIFG has been widely investigated as a core node of the inhibitory control network[49,50], though its specific role in the stopping process is still under debate[51,52]. A reactive global stop model has proposed that rIFG may send the primary input to the STN via the hyperdirect pathway[53]. In the present study children showed a strong effect in the rAI-STN pathway such that stronger task-related modulation was correlated with faster stopping action with a trend in the same direction for task-evoked interaction between rIFG and rSTN (Supplementary Table 10). Thus, both prefrontal regions may influence the STN.

In conclusion, inhibitory control is fundamental to cognitive and affective development in childhood. We found that by late childhood, children can recruit an adult-like inhibitory control brain network during successful stopping. We found that the degree to which a child's brain elicits an adult-like global activation pattern predicts their inhibitory control abilities. Furthermore, we found that the degree of functional response in right STN, a key node in the hyperdirect pathway, and its functional connections with the right AI, underlie individual differences in inhibitory control ability in children. Crucially, identification of the STN and its cortical circuitry as an important locus for understanding the development of inhibitory control in children moves us away from the cortico-centric view that has persisted thus far. In an advance over previous studies, these key findings were replicated across two different cohorts of children, including participants from the ABCD study. Our study provides insights into cortical–subcortical mechanisms underlying immature inhibitory control in the developing brain, and establishes a template for investigating inhibitory control deficits in neurodevelopmental disorders such as ADHD and autism. Our analytic approach and the NMI examined here may be relevant for other neurodevelopment studies and may help further facilitate the building of deeper links between seemingly disparate open-source neuroimaging datasets.

## Methods

**Ethics statement**. Acquisition of the Stanford_Child dataset was approved by Institutional Review Board at Stanford University. OpenfMRI_Adult1 and OpenfMRI_Adult2 datasets were approved by the University of California, Los Angeles' Institutional Review Board. The ABCD study was reviewed and approved by the University of California, San Diego's Institutional Review Board. All the participants have provided written consents.

**Meta-analysis of inhibitory control studies in children**. We conducted an activation likelihood estimation (ALE)[54] to investigate common brain mechanisms of inhibitory control during mid/late childhood in the summer of 2018. Studies included in the meta-analysis were required to meet the following criteria: (1) study included children between 6 and 13 years of age; (2) study reported activation peaks in typically developing children, distinct from adult or clinical groups if any; (3) study used the SST or the GNGT; (4) study included activation contrast analysis that directly probes inhibitory control; (5) study reported activation peaks using whole-brain analysis; and (6) the activations were reported in either Montreal Neurological Institute (MNI) or Talairach space. Literature search was conducted by early November of 2017 using PubMed (www.ncbi.nlm.nih.gov/pubmed/) and key words like fMRI, SST, Go/NoGo task and children. Using the inclusion criteria noted above, 14 studies were selected for inclusion in the meta-analysis (Supplementary Table 1).

Meta-analyses were conducted using GingerALE (http://www.brainmap.org/ale/)[54,55]. First, we selected all the contrasts that probed inhibitory control in each study entered in the meta-analysis. Activation coordinates from all selected contrasts were converted into MNI space. Each coordinate was modeled by a 3D Gaussian distribution and the ALE value for each voxel in the brain was calculated. Next, a permutation procedure was applied to create a null distribution of the ALE value at each voxel from which the $p$ value of ALE at each voxel was computed. Lastly, an activation threshold was applied to generate an output map ($p < 0.01$, FWE corrected, 5000 permutations). However, no significant clusters survived in this meta-analysis. For the purpose of visualization ($p < 0.01$, uncorrected), we lowered the threshold to compare the overall brain activation pattern in children to adults.

Because we were interested in examining similarity in brain activation patterns during inhibitory control between children and adults, we reanalyzed data from a previous meta-analysis[17], using the latest GingerALE version 2.3.6. The same threshold was applied ($p < 0.01$, FWE corrected, 5000 permutations).

**Adult SST OpenfMRI datasets**. Two adult SST fMRI datasets were obtained from a public fMRI database—OpenfMRI (http://openfmri.org). We labeled the two datasets as Adults_dataset_1, which is ds7 from OpenfMRI.org[16] and OpenfMRI_Adult2, which is ds9 from OpenfMRI.org.

In the OpenfMRI_Adult1[16], 18 subjects (18–39 years old) made button-press responses to a letter T or D (Go signal). In 25% of trials, the letter was followed by a beep (Stop signal) and subjects attempted to stop their responses. The stop-signal delay (SSD) in the practice session was determined using a stepwise procedure. Eight SSDs were generated based on the average SSD in the practice session ±60/20 ms and used in the fMRI session. Each participant completed two runs of SST, including 128 trials per run.

In the OpenfMRI_Adult2, 24 subjects (18–33 years old) made button-press responses to a leftward or rightward pointing arrow (Go signal), which was occasionally (about 25% of trials) followed by a tone (Stop signal). If the stop signal was presented, subjects had to stop their responses. The SSD varied dynamically in a staircase procedure. If the subject successfully stopped on a stop trial, the SSD increased by 50 ms on the next stop trial. If the subject failed to stop on a stop trial, the SSD decreased by 50 ms on the next stop trial. Each participant completed two runs of SST, including 128 trials per run.

**Stanford_Child SST dataset**. The child SST fMRI dataset has not been previously published. It included 38 subjects (12 female, all right handed with no history of neurological or psychiatric disorders, 9–12 years of age). Other participant information is reported in Supplementary Table 5.

Each child performed two runs of the SST, including 96 trials per run. Subjects were instructed to respond as quickly as possible to green arrows (Go Signal) by clicking with their right pointer or middle finger based on the direction of the arrow. In 33% of the trials, after a variable delay, the green arrow turned red (Stop Signal), indicating that the subject should cancel their response. The delay between the Go Signal and the Stop Signal, the SSD varied across trials in a step-wise fashion and adjusted dynamically to the subject's performance: beginning at 165 ms, it decreased by 33 ms for a failed stop, and increased by 33 ms for a successful stop.

The fMRI data were acquired on a 3T GE Signa scanner using an 8-channel head coil. Functional images of 29 axial slices, parallel to the anterior/posterior commissure line and covering the whole brain, were acquired using a T2*-weighted gradient-echo spiral in–out pulse sequence with the following parameters: slice-thickness = 4.0 mm, repetition time (TR) = 2000 ms, echo time (TE) = 30 ms, flip angle = 80°.

For participants, task, and MRI protocol details, see Supplementary Methods.

**ABCD replication dataset**. The ABCD study (https://abcdstudy.org) includes a similar SST in the fMRI data[37]. We analyzed SST fMRI data from the first 500 typically developing children (TDC) aged 9–11 to replicate the major findings from the Stanford_Child SST dataset. For participants details, see Supplementary Methods.

Each child completed two runs of the SST, containing 180 trials. In each trial, children made responses to a leftward or rightward pointing arrow (Go signal). Occasionally (16.67%), the leftward or rightward arrow was followed by an up-right arrow (Stop signal). Children were instructed to withhold responses for the stop signal. The initial SSD was 50 ms and its value varied based on performance: SSD reduced by 50 ms after an unsuccessful stop trial and increased by 50 ms after a successful stop trial.

Further task and protocol details can be found in the ABCD study[37].

**fMRI preprocessing**. FMRI data were preprocessed using SPM8, including realignment, slice-timing correction, co-registration, normalization, and smoothing. For details, see Supplementary Methods.

**Task fMRI GLM analysis**. A general linear model (GLM) analysis was used to model task-related effects. Six motion parameters were entered as covariates of no interest, and task-evoked hemodynamic response was calculated at each voxel using the canonical hemodynamic response function for the four conditions of interest: Go Correct (Go), Go Error, Successful Stop (SuccStop), and Unsuccessful Stop (UnsuccStop). Contrasts of interest (SuccStop versus Go and SuccStop versus UnsuccStop) were generated and used to create voxelwise $t$-statistics maps for each subject. Group-level activation statistic maps were thresholded at $p < 0.01$, FDR corrected.

We used the contrast SuccStop vs. Go to examine brain activations and similarity between children and adult and to develop an NMI because it best captures neural substrates underlying stimulus-triggered stopping, including signal detection, triggering the stop process, and action cancellation.

**Similarity analysis of activation during inhibitory control**. We quantitatively assessed the similarity in brain-wide activation patterns during inhibitory control between children and adults. First, we computed *Pearson's* correlations of contrast images (e.g. SuccStop—Go) between child and adult cohorts across all the voxels in gray matter except sensorimotor cortex. The sensorimotor cortex was excluded because the neural response pattern from these voxels are largely dependent on the specific choice of sensory stimuli and motor response in a study. Therefore, including those voxels may over-estimate or under-estimate the similarity of activation pattern underlying inhibitory action control. The significance of brain-wide spatial correlations was estimated using permutation testing. Specifically, in each permutation, we randomly shuffled voxels in adult brain activation map. We then computed the correlation between the shuffled maps and children activation maps. We repeated the permutation procedure 100 times with a different random seed each time and obtained a random distribution of spatial correlation between two brain activation maps from which the $p$-value was computed. Second, a similar analysis was conducted using an independent set of whole-brain ROIs derived from a previous study[40]. For the same reason, we excluded ROIs from sensory and motor networks. The brain networks included in the analysis were the salience network (SN), subcortical network (SN), default mode network (DMN), dorsal attention network (DAN), VAN, FPN, cingulo-opercular network (CON) and memory network (Mem).

**Neural maturity index**. To quantify to what extent children's neural responses across the brain during inhibitory control are adult-like or mature, we developed a NMI[56]. Specifically, we first obtained group-level $t$-stats map for the contrast of interest (SuccStop—Go) from each of the two adult datasets, labeled as an 'adult reference map', and created a mask using significant activation and deactivation voxels in the adult reference map ($p < 0.01$, FDR corrected). Then, we computed the spatial correlation between each child's subject-level $t$-stats map and the adult reference map in the mask, which represents activation/deactivation similarity between a child subject and the adult group. Finally, the child's NMI was defined as the Fisher transformed correlation coefficient of the child's dataset. Note that a child's NMI value could change if a different adult reference map was used. To examine the robustness of the NMI measure, we used two different adult fMRI datasets to demonstrate reproducibility of our findings. One child's NMI map was identified as an outlier with a value that was three standard deviations away from the group mean and was removed from the following analyses.

**ROI analysis**. To compose an inhibitory control brain network, we selected top activation peaks from a previous meta-analysis of adult studies[17], including the rAI, rIFG, rMFG, rPreSMA, rSMG and rCau. Each ROI was constructed as a sphere of 6 mm radius using Marsbar (http://marsbar.sourceforge.net/). In addition, we obtained coordinates of the bilateral STN (lSTN and rSTN), key nodes of the hyperdirect pathway, from a previous ultra-high MRI study of the STN and built the ROIs as a sphere of 4 mm radius[28]. Importantly, we verified the location of STN mask on each individual participant. Coordinates of all the ROIs were summarized in Supplementary Table 8.

**Classification analysis**. We examined whether voxel-wise activation pattern within STN ROIs could differentiate between Go and SuccStop. To do so, we applied multivariate classification using the linear support vector machine algorithm ($C = 1$) and a leave-one-subject-out cross-validation procedure. For details, see Supplementary Methods.

**Brain–behavior correlation analysis**. Correlation analysis was conducted to examine the relationship between task-evoked activation in ROIs and SSRT. Beta values were extracted from each ROI for the condition of interest (SuccStop > Go) for each participant and then correlated with participants' SSRT. Multiple linear regression was used to examine whether brain–behavior relationships could be driven by potential confounds, such as age, gender, and head motion[57].

**Effective connectivity and behavioral correlation analysis**. Task-modulated effective connectivity was computed using seed-based generalized psychophysiological interactions method (gPPI)[58]. Seeds were selectively placed in the rAI, rIFC, rMFG, and rPreSMA, as we were testing a hyperdirect pathway model, which projects from cortical regions to the STN. For gPPI details, see Supplementary Methods. Based on the Race model[29], whether a prepotent response can be canceled successfully depends on which process finishes at first. Therefore, the SuccStop and UnsuccStop conditions share some early stopping-related processes, such as signal detection and triggering the stop process, which allows us to focus on differences in later stages of the stop process, such as action cancellation. The focus of our connectivity analyses was to test the hypothesis that the hyperdirect pathway linking the STN and the cortex is involved in this later stage of the stopping process. Therefore, we specifically focused on whether task-evoked interactions between cortical regions and STN in SuccStop versus UnsuccStop are associated with individual abilities to stop prepotent responses. We computed task-modulated effective connectivity of the STN and computed its correlation with individual's SSRT using Pearson's correlations. Multiple linear regression was used to examine whether any brain–behavior relationships could be driven by other potential confounds, such as age, gender, and head motion.

**Reporting summary**. Further information on research design is available in the Nature Research Reporting Summary linked to this article.

## Data availability
The data used in this study are available from the authors upon request. A reporting summary for this Article is available as a Supplementary Information file. Source data underlying Figs. 3–7, Tables 2–4, Supplementary Figs. S1–9 and Supplementary Tables S10–23 are provided as Source Data files.

## Code availability
The code used in this study is available on our lab website (https://med.stanford.edu/scsnl/publications.html).

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

## Acknowledgements

This work was supported by the Lucile Packard Foundation for Children's Health (V.C., V.M.) and National Institutes of Health grants EB022907 (V.M.), NS086085 (V.M.), MH121069 (V.M., W.C.) and MH105625 (W.C.). We thank Dr. Shaozheng Qin, Sarah Nicole Bostan, Olivia Altamirano, and Yamilka Alsina for their assistance with data collection. We thank the Poldrack Lab and Center for Reproducible Neuroscience at Stanford University for managing the OpenfMRI database. Data used in replication analysis were obtained from the Adolescent Brain Cognitive Development (ABCD) study (https://abcdstudy.org/). We thank Carlo de los Angeles for his assistance with downloading data from the ABCD study.

## Author contributions

Conceptualization and Design: W.C., V.C., V.M.; Data Acquisition: W.C., K.D., A.P., R.R., T.B.; Methodology: W.C., V.M.; Data Analysis: W.C., K.D.; Writing: W.C., A.P., V.M.; Review and editing: W.C., K.D., A.P., V.C., V.M.

## Competing interests

The authors declare no competing interests.
