## [Peer Review File · Nature Communications]

Reviewers' Comments:

Reviewer #1:

Remarks to the Author:

In this manuscript, the authors report on multiple analyses collectively aimed at understanding inhibitory control in children. Specifically, focus was placed on examining the role of cortical-basal ganglia pathways during stopping behavior in children and the extent to which children show mature, "adult-like" responses. First, the authors conducted meta-analyses to better understand how activation patterns reported in previous studies in children aligned with those in adults. Next, the authors used event related fMRI datasets to assess adult-like maturity of inhibitory control in the SST, focusing on similarity of brain activation patterns between children and adults. From this, the authors derived a neural maturity index as a means to quantify activation similarity between age groups. Finally, the authors investigated the hyperdirect pathway in children by examining task-induced activation in the STN and SSRT and task-related effective connectivity of the STN and anterior insula.

Overall, the analyses were well reasoned and the authors presented cogent evidence in support of their aims. Notably, the paper was very well written – clear yet appropriately detailed to enable replication.

Below is a list of minor comments/questions noted during my reading of this manuscript:

1. In the description of the Stanford_Child dataset found in the Supplementary Information, it would be useful to include, if possible, additional participant information like IQ, SES, if any participants were on stimulant medication for ADHD, etc., so that the reader can get a better sense of the generalizability of data from this sample. A note on generalizability may also be appropriate in the discussion. If these data are not available, then mentioning that would also be worthwhile.
2. It was unclear to me how many trials or volumes were included in the event related analyses? Could the authors please clarify?
3. To clarify, in the data plotted in Figure 3 panel D, do the x and y label represent beta values from the SuccStop vs Go contrast? Please indicate label in the figs or caption. Also, what would these plots look like if there was a control dataset (i.e., a dataset from adults performing a different task or showing a different contrast – like "go" trials) plotted against the Stanford_Child data? In other words, how much of this correlation comes from more general "task positive" type activation?
4. I realize it's not the focus of the manuscript here, but it would be extremely interesting to also see similarity/differences between the children and adult datasets in terms of error trial activation. (This is not a request for additional analyses, just a comment!)
5. In the analysis examining NMI and SSRT (e.g., Fig 6 panel B) were there any outliers in the reaction times and, if so, were they removed?
6. Was there any indication of differences in brain sizes (volume) between children and adults? Could this affect spatial comparisons?

Reviewer #2:

Remarks to the Author:

This is a very useful and timely meta-analysis of data from event-related fMRI studies of inhibitory response control in children of 9-12 years of age, with additional analysis of three additional and presumed novel, Stanford data-sets - to provide a valuable perspective on the neurodevelopment of human inhibitory control.

The authors confirm the canonical network for the stop-signal task (SST) in adults and show, with some difficulty from the few relevant published studies, a similar network in children. This is confirmed however by analyses of additional novel fMRI data-sets, together with a strong quantitative relation between children and adults at the brain-wide level, despite quantitative differences in behavioral indices of SST. 9-12 year old children are also shown to have adult-like brain activation in cognitive control systems (i.e. the salience, frontoparietal, ventral attentional and default mode networks). The comparison with adult networks is partly based on the use of a novel index of a "neural maturation index" (NMI; derived from the child functional imaging data in relation to two adult data-sets) that can be used to characterize individual differences in development. Next, the authors focus on the role of the subthalamic nucleus (STN) in this network, including the hyper-direct pathway from the cortex, a theoretically controversial component of the 'stopping network'. They find that the rostral STN can distinguish Go from successful stopping trials in children and that there is a negative correlation between STN activation and inhibitory control, as previously published in adults. At the cortical level, STN activation is associated with the anterior insula in children (but not in adults) and predicts inhibitory control ability.

Overall, I found this to be a comprehensive and rigorous intellectual (including statistical) analysis which does much to illuminate the neural development of this aspect of inhibitory function, relevant to range of developmental disorders, such as ADHD, autism (and OCD) - an impressive achievement that deserves publication in a high impact journal. I did have a few relatively minor specific points for the author's consideration, that might be addressed in a revision, as listed below:

Specific points:

1. The text is commendably succinct, though at times dense to read and somewhat relentless! Despite this, it is generally clear except in a few places. For example, the sentence on p5 about conforming or violating the race model could be clarified: "such that RT in unsuccessful stop trials is shorter than RT in Go trials" - Please re-phrase; the sentence could be read as ambiguous.
2. Can it be made explicit whether the analyses of the Stanford datasets were of novel fMRI data and not previously published? (I assume so; though it is not a crucial issue).
3. Can the behavioral data be used to provide a "Functional Maturity Index" for comparison with the adult data? Are the behavioral data, in fact, more sensitive to development than the neural network differences, as reflected by the NMI?
4. There was no apparent effect on anterior insula-STN connectivity in adults, contrasting with children- is that the case? Please clarify. It is one of the major differences between the two data-sets, but its functional and developmental significance is not considered much further in the Discussion. The findings appear to vary from the imaging data and hypothesis of Poldrack and Aron - perhaps this needs to be pointed out in relation also to your earlier meta-analysis of adult data in the Journal of Neuroscience. Is there evidence that the insula-STN connectivity is statistically greater than that of the right IFG-STN connectivity, in either children or adults? This would help to provide evidence of the presumed specificity of the insula-STN hyper-direct pathway.
4. The relevance of the parallel systems of cognitive control during fMRI of SST is not made transparent in the Discussion.
5. The opportunity to provide scholarly citations should perhaps be augmented, although I realise there may be constraints of space. The data are highly relevant to the theoretical articles published in TICs by Aron et al 2004, 2014. The fMRI-SST paper by Whelan et al in Nature Neuroscience, though of 14year olds, is also of clear relevance to this developmental picture.

Reviewer #3:

Remarks to the Author:

This study shows that, during stop-signal task performance, children age 9–12 ($n = 38$) with more “adult-like” activity patterns show better inhibitory control performance. In addition, subthalamic nucleus activity and functional connectivity is related to inhibitory control in a developmental sample, but not in two adult samples ($n_s = 18$ and 24). Although this study tackles an extremely interesting question with a clear, logical and multi-faceted approach, my enthusiasm is tempered by the relatively small developmental sample, which makes it difficult to disentangle developmental effects from individual differences. Additional replication data to support the claims that the “neural maturation index” reflects brain maturity *per se* and that the STN is involved in inhibitory control in development would significantly strengthen the manuscript.

Major comments:

1) The small developmental sample and age range make it difficult to dissociate developmental effects from individual differences. Complementary approaches for disentangling these possibilities including testing a larger developmental sample and age range to show developmental change in inhibitory control (using, for example, Adolescent Brain Cognitive Development, IMAGEN, or PING data with a different measure of inhibitory control), and testing whether adults closer to the adult average template also show better SSRTs (this finding would suggest individual differences rather than developmental change is driving the result). Including data with other measures of inhibitory control could be particularly informative as previous work suggests that age only accounts for about 11% of the variance in SSRT (Bedard et al. 2002, *Developmental Neuropsychology*).

2) The “neural maturation index” is defined as the spatial correlation between each child’s activation map and the average adult map in a mask defined in the adult data. This approach leaves open the possibility that children with higher NMIs actually show less similar patterns at the whole-brain level. Do results hold for whole-brain rather than ROI-based correlations?

3) Does the NMI reflect state-like or trait-like effects? The name suggests that it reflects trait-like developmental effects, but it is possible that it may reflect state-like effects such that trials/runs with good performance show more “typical” adult-like neural pattern and less successful trials/run show less “typical” patterns. Would this result change the authors’ interpretation of the results?

4) In the developmental dataset, the motion exclusion thresholds of >5 mm displacement and $>.5$ mm average frame-to-frame displacement (is that what’s meant by “scan-to-scan movement”?) are high compared to previous studies, which typically exclude runs with $>2-3$ mm maximum displacement and/or $>.15-.3$ mm mean frame-to-frame displacement. Are connectivity results consistent in individuals with low head motion, and consistent when controlling for mean (rather than maximum) framewise displacement?

5) Were decisions about which contrasts to analyze made before analyses were performed? The motivation to report results from the successful stop vs. successful go contrast for activation and classification analyses, but the more-common successful vs. unsuccessful stop contrast for functional connectivity analysis, is unclear.

6) The STN results, while interesting, are relatively weak. Results in the two adult datasets do not replicate previous findings or the current developmental finding that SSRTs are inversely correlated with activation in STN activation during inhibitory control. STN classification and connectivity results are also not reported for adult samples. Replication in a larger developmental sample would increase confidence that (a) the STN was appropriately localized in children and (b) classification and individual differences results are robust.

Minor comments:

- 1) The go/no-go task quantifies accuracy but not SSRT, whereas the stop-signal task quantifies SSRT but not accuracy (because task timing is individualized). A more accurate characterization of the tasks is that both have benefits and drawbacks for characterizing inhibitory control.
- 2) Why does the meta-analytic finding suggest that, "in contrast to adults, extant functional neuroimaging studies of inhibitory control in children are less consistent"? Could this be driven by the inclusion of both SSTs and go/no-go tasks in the developmental meta-analysis?
- 3) It would be helpful for the authors to unpack what they mean by a "hyperdirect" pathway, which is a major focus of the text but never fully explained. In other words, what distinguishes a direct pathway from a hyperdirect pathway?
- 4) Correlational analyses should be clearly distinguished from predictive analyses. The term "predict" should be reserved for cross-validated models.
- 5) In Figure 4D, it appears that significance is determined by parametrically converting r values to p values. However, data points (voxels) are non-independent, so the degrees of freedom are likely overestimated, compromising the validity of this conversion.
- 6) It would be helpful to keep the bar graph axes consistent in Figure 4C.
- 7) In table S7 it would be helpful to include the number of voxels per ROI in the table.
- 8) Rather than making the code available upon request it would be helpful to share the code in a publicly available repository (e.g., github) along with links to the publicly available datasets analyzed.

Reviewer #4:

Remarks to the Author:

The goal of this study was to examine the neural mechanisms of inhibitory control using multiple approaches (meta analysis, open-source fMRI datasets, primary analysis). The use of multi-level approaches is a strength of the study. They found that by and large children recruit the same brain regions as adults during stopping. They also calculated a "neural maturity index". The methods are sound, the driving questions are rooted in a strong theoretical foundation, and the conclusions are appropriate. However, my enthusiasm for this manuscript was dampened by the lack of clarity regarding the impact on the field. It is unclear how this study provides "new insights" about immature inhibitory control in the developing brain. Identification of the STN as important for inhibitory control in children is interesting but not transformative. Some specific comments below:

- 1) The meta-analysis is important but it is unclear how the inclusion criteria of 6-13 years of age was chosen. This broad age range includes adolescents and individuals undergoing puberty, both of which confound understanding of inhibitory control in children.
- 2) The primary data analysis of the Child SST contains a fairly small sample size at $n=38$. What were the effect sizes?
Why were only 38 of the 78 participants who were recruited for the larger study included in this analysis?
- 3) A motion threshold of 5mm is quite high--with the norm being between 2-3mm in the field. How much data was lost to motion? and how many participants had greater than 3mm of motion

displacement?

4) It is surprising that the authors only included two studies of the SST in the analysis, as several other groups have published fMRI studies of SST in children. Admittedly, many of these other manuscripts included other populations of children as well but the healthy comparison groups could have been included here.

5) The adult reference group data for the NMI calculation were derived from the two OpenfMRI adult datasets, which themselves contain a broad age range of participants, spanning 18-39 years and 18-33 years, respectively. This is problematic because of the significant neural changes occurring during the ages of 18 to roughly 25/26 years of age, which makes use of these datasets as 'adult reference groups' concerning.

Reviewer #1

1.1 *“In this manuscript, the authors report on multiple analyses collectively aimed at understanding inhibitory control in children. Specifically, focus was placed on examining the role of cortical-basal ganglia pathways during stopping behavior in children and the extent to which children show mature, “adult-like” responses. First, the authors conducted meta-analyses to better understand how activation patterns reported in previous studies in children aligned with those in adults. Next, the authors used event related fMRI datasets to assess adult-like maturity of inhibitory control in the SST, focusing on similarity of brain activation patterns between children and adults. From this, the authors derived a neural maturity index as a means to quantify activation similarity between age groups. Finally, the authors investigated the hyperdirect pathway in children by examining task-induced activation in the STN and SSRT and task-related effective connectivity of the STN and anterior insula.*

Overall, the analyses were well reasoned and the authors presented cogent evidence in support of their aims. Notably, the paper was very well written – clear yet appropriately detailed to enable replication.”

Response: We are pleased that the reviewer found our work to be well reasoned, and well written in a manner that will facilitate replication. Please note that data analysis scripts used in this study will be made available upon publication.

1.2 *“ In the description of the Stanford_Child dataset found in the Supplementary Information, it would be useful to include, if possible, additional participant information like IQ, SES, if any participants were on stimulant medication for ADHD, etc., so that the reader can get a better sense of the generalizability of data from this sample. A note on generalizability may also be appropriate in the discussion. If these data are not available, then mentioning that would also be worthwhile.”*

Response: We thank the reviewer for this suggestion. We acquired a number of neuropsychological measures, including Kaufman Assessment Battery for Children which assesses IQ, Behavior Rating Inventory of Executive Function, Behavior Assessment System for Children and SES scores which have been added to the revised manuscript (**SI Table S5**). We have also included a sentence about generalizability in the discussion section as noted below:

*“Importantly, the distribution of behavioral and cognitive-affective scores were in the normative range of neuropsychological function as assessed using multiple measures including the Kaufman Assessment Battery for Children which measures IQ and achievement, Behavior Rating Inventory of Executive Function and Behavior Assessment System for Children which assesses behavioral and emotional strengths (**SI Table S5**). This suggests that in a sample that is representative of typically developing children ages 9-12 years, inhibitory control mechanisms are not fully mature by late childhood.”*

1.3 *“It was unclear to me how many trials or volumes were included in the event related analyses? Could the authors please clarify?”*

Response: We apologize that the number of trials was not clear in the manuscript. This information was included in the Supplementary Information in the original submission, but we

have moved it to the main methods section of the manuscript in this revision. In the Stanford_Child dataset, each child completed two runs of the Stop-Signal task, including 96 trials per run. In the OpenfMRI_Adult1 and OpenfMRI_Adult2, each participant completed two runs of the stop-signal task, including 128 trials per run. (see pages 14-15)

1.4 “To clarify, in the data plotted in Figure 3 panel D, do the x and y label represent beta values from the SuccStop vs Go contrast? Please indicate label in the figs or caption. Also, what would these plots look like if there was a control dataset (i.e., a dataset from adults performing a different task or showing a different contrast – like “go” trials) plotted against the Stanford_Child data? In other words, how much of this correlation comes from more general “task positive” type activation?”

Response: We have revised the figures and captions to make them clear as suggested.

While general “task positive” activation does contribute to the correlation between Child and Adult datasets in this analysis, there is also a clear spatial similarity in the voxels that have negative beta values between Child and Adult datasets (**Figure 3D**).

Moreover, we conducted additional analyses to examine spatial correlation during Go trials between Child and Adult datasets. While the spatial activation pattern during Go trials was correlated between Stanford_Child and Adult_dataset_1 ($r=0.36$, **SI Figure S1**) and between Stanford_Child and Adult dataset_2 ($r=0.41$, **SI Figure S1**), the correlation coefficient was much lower in comparison to spatial correlation during stopping between Child and Adult datasets ($r=0.69$ and $r=0.71$, **Figure 3D**). We have included this analysis in the Supplementary Materials. Taken together, these analyses suggest that the correlation between activation patterns elicited by children and adults during stopping is not solely due to general task positive type activation.

Figure S1. Brain-wide activation patterns elicited by motor execution (Go trials) in Children were correlated with activation patterns in the two Adult cohorts. Each data point represents one voxel’s beta contrast of Go in gray matter mask.

1.5 “I realize it’s not the focus of the manuscript here, but it would be extremely interesting to also see similarity/differences between the children and adult datasets in terms of error trial activation. (This is not a request for additional analyses, just a comment!) “

Response: We thank the reviewer for this excellent suggestion and agree that it would be interesting to conduct the suggested analysis. To address this comment, we conducted additional analysis to examine spatial correlation related to error trial activation with the contrast UnsuccStop versus SuccStop between Child and Adult datasets. Error-elicited spatial activation patterns were correlated between Stanford_Child and Adult_datset_1 ($r=0.43$, **SI Figure S2**) and between Stanford_Child and Adult_datset_2 ($r=0.31$, **SI Figure S2**), but the correlation coefficient was much lower in comparison to spatial correlation during stopping between Child and Adult datasets ($r=0.69$ and $r=0.71$, **Figure 3D**). We have included this analysis in the Supplementary Materials.

Figure S2. Brain-wide activation patterns elicited by errors in children were related to activation patterns in two different cohorts of adult. Each of the data points represents one voxel's beta contrast of UnsuccStop versus SuccStop in grey matter mask.

1.6 “In the analysis examining NMI and SSRT (e.g., Fig 6 panel B) were there any outliers in the reaction times and, if so, were they removed?”

Response: We thank the reviewer for attention to this detail. There were no outliers in the SSRT measures. However, there was one NMI outlier, which was defined as more than 3 standard deviations away from the group mean. This outlier was removed from the NMI and behavioral correlation analysis. We have made this clear in the revised manuscript. (see page 16)

1.7 “Was there any indication of differences in brain sizes (volume) between children and adults? Could this affect spatial comparisons?”

Response: We agree with the reviewer that brain size differences may have an impact on spatial comparison between children and adults. To address any potential effects of brain size, we used non-linear registration to spatially normalize child and adult brains into a standard template. This is a standard preprocessing step in fMRI studies particularly for individuals ages 9 and above.

Reviewer #2

2.1 *“This is a very useful and timely meta-analysis of data from event-related fMRI studies of inhibitory response control in children of 9-12 years of age, with additional analysis of three additional and presumed novel, Stanford data-sets - to provide a valuable perspective on the neurodevelopment of human inhibitory control.*

The authors confirm the canonical network for the stop-signal task (SST) in adults and show, with some difficulty from the few relevant published studies, a similar network in children. This is confirmed however by analyses of additional novel fMRI data-sets, together with a strong quantitative relation between children and adults at the brain-wide level, despite quantitative differences in behavioral indices of SST. 9-12 year old children are also shown to have adult-like brain activation in cognitive control systems (i.e. the salience, frontoparietal, ventral attentional and default mode networks). The comparison with adult networks is partly based on the use of a novel index of a "neural maturation index" (NMI; derived from the child functional imaging data in relation to two adult data-sets) that can be used to characterize individual differences in development. Next, the authors focus on the role of the subthalamic nucleus (STN) in this network, including the hyper-direct pathway from the cortex, a theoretically controversial component of the 'stopping network'. They find that the rostral STN can distinguish Go from successful stopping trials in children and that there is a negative correlation between STN activation and inhibitory control, as previously published in adults. At the cortical level, STN activation is associated with the anterior insula in children (but not in adults) and predicts inhibitory control ability.

Overall, I found this to be a comprehensive and rigorous intellectual (including statistical) analysis which does much to illuminate the neural development of this aspect of inhibitory function, relevant to range of developmental disorders, such as ADHD, autism (and OCD) - an impressive achievement that deserves publication in a high impact journal.”

Response: We are pleased that the reviewer found our work to be comprehensive, intellectually rigorous, and impressive. We hope our methods will serve as a template for future neurodevelopmental studies.

2.2. *“The text is commendably succinct, though at times dense to read and somewhat relentless! Despite this, it is generally clear except in a few places. For example, the sentence on p5 about conforming or violating the race model could be clarified: "such that RT in unsuccessful stop trials is shorter than RT in Go trials" - Please re-phrase; the sentence could be read as ambiguous.”*

Response: We thank the reviewer for this suggestion. Accordingly, we have modified the sentence to “For example, a key requirement here is that Unsuccessful Stop trials should have shorter RTs than Go trials” in the results section of the revised manuscript and carefully gone over the text to correct any other possibly ambiguous statements.

2.3. *“Can it be made explicit whether the analyses of the Stanford datasets were of novel fMRI data and not previously published? (I assume so; though it is not a crucial issue).”*

Response: Yes, the Stanford_Child cohort is a novel fMRI dataset that has not previously published. We have made this explicit in the revised manuscript. (see page 15)

2.4. *“Can the behavioral data be used to provide a “Functional Maturity Index” for comparison with the adult data? Are the behavioral data, in fact, more sensitive to development than the neural network differences, as reflected by the NMI?”*

Response: We thank the reviewer for this comment. We used similarity of brain activation patterns between children and adults as a measure of functional maturity (i.e. the NMI) and then validated the NMI by its correlation with behavioral measures (e.g. SSRT). In principle, it is possible to develop a similar distance metric based on behavioral or neuropsychological scores. However, because we do not have other measures related to cognitive control in both children and adults (unlike the Stanford_Child dataset, the two Open fMRI datasets from adults contained no neuropsychological assessments), we could not derive behavioral maturational indices.

2.5. *“There was no apparent effect on anterior insula-STN connectivity in adults, contrasting with children- is that the case? Please clarify. It is one of the major differences between the two data-sets, but its functional and developmental significance is not considered much further in the Discussion. The findings appear to vary from the imaging data and hypothesis of Poldrack and Aron - perhaps this needs to be pointed out in relation also to your earlier meta-analysis of adult data in the Journal of Neuroscience. Is there evidence that the insula-STN connectivity is statistically greater than that of the right IFG-STN connectivity, in either children or adults? This would help to provide evidence of the presumed specificity of the insula-STN hyper-direct pathway.”*

Response: We thank the reviewer for this thoughtful comment. We found that effective connectivity of rAI-rSTN is significantly correlated with SSRT in our Child cohort but not in the two independent Adult cohorts. To our knowledge no previous studies have reported an association with cortical-rSTN connectivity and SSRT in adults. Little has been known about the relationship between Stop signal-modulated cortical-STN effective connectivity and Stopping ability in either adults or in children.

gPPI analysis revealed that Stop Signal-modulated connectivity between the rAI and rSTN, but not the ISTN, during stopping was significantly correlated with SSRT ($r=-0.46$, $p<0.005$, *Cohen’s* $d=1.04$) in children, but not in adults. Moreover, Stop Signal-modulated connectivity between the rIFG and rSTN was marginally significantly correlated with SSRT in children ($r=-0.30$, $p=0.07$, *Cohen’s* $d=0.63$), but not in adults.

To address the issues raised by the reviewer, we investigated task-evoked effective connectivity of IFG-STN and AI-STN circuits during stopping in relation to individual differences in SSRT in children and adults. To provide a full picture of our gPPI analysis, we have now reported all the results of correlation between the cortical-STN gPPI strength and SSRT in Stanford_Child (**SI Table S10**), OpenfMRI_Adult1 (**SI Table S11**) and OpenfMRI_Adult2 (**SI Table S12**) datasets in revised manuscript.

Finally, to address the question raised by the reviewer, we examined whether right AI-STN connectivity is statistically greater than that of the right IFG-STN connectivity. In both children and adults, we found no significant difference in gPPI weights between the right AI-STN and IFG-STN connectivity ($ps>0.1$).

At present it is not completely clear why a significant correlation between rAI-STN and SSRT emerged in children but not in adults. One possibility is that children may be more dependent on Stop-signal driven modulation of the hyperdirect cortical-STN pathway because of immature proactive and anticipatory control mechanisms. Adults, on the other hand, may better implement proactive control mechanisms, which is less dependent on the hyperdirect cortical-STN pathway. Further research is required to test this hypothesis. We have updated the text accordingly.

Table S10. Correlation between effective connectivity between cortical seeds and STN target and SSRT in Stanford_Child dataset.

seed	target	r	p
rAI	ISTN	-0.03	0.87
	rSTN	-0.46	0.004
rIFG	ISTN	-0.16	0.32
	rSTN	-0.3	0.07
rPreSMA	ISTN	-0.02	0.94
	rSTN	-0.25	0.13
rMFG	ISTN	-0.24	0.16
	rSTN	-0.02	0.92

Table S11. Correlation between effective connectivity between cortical seeds and STN target and SSRT in OpenfMRI_Adult1 dataset.

seed	target	r	p
rAI	ISTN	0.19	0.45
	rSTN	-0.11	0.65
rIFG	ISTN	0.01	0.99
	rSTN	-0.03	0.89
rPreSMA	ISTN	-0.07	0.78
	rSTN	-0.55	0.02
rMFG	ISTN	-0.02	0.92
	rSTN	-0.07	0.79

Table S12. Correlation between effective connectivity between cortical seeds and STN target and SSRT in OpenfMRI_Adult2 dataset.

seed	target	r	p
rAI	ISTN	0.08	0.71
	rSTN	0.23	0.3
rIFG	ISTN	-0.14	0.53
	rSTN	-0.11	0.6
rPreSMA	ISTN	-0.17	0.43
	rSTN	-0.08	0.71
rMFG	ISTN	-0.11	0.63

rSTN	0.07	0.76
------	------	------

2.6. *“The relevance of the parallel systems of cognitive control during fMRI of SST is not made transparent in the Discussion.”*

Response: We thank the reviewer for this comment. We have revised the manuscript and include discussion of the parallel systems of cognitive control during fMRI of SST (see pages 12-13).

2.7. *“The opportunity to provide scholarly citations should perhaps be augmented, although I realise there may be constraints of space. The data are highly relevant to the theoretical articles published in TICs by Aron et al 2004, 2014. The fMRI-SST paper by Whelan et al in Nature Neuroscience, though of 14year olds, is also of clear relevance to this developmental picture.”*

Response: We thank the reviewer for reminding us of these studies. We have cited and discussed these studies in our revised manuscript.

Reviewer #3

3.1 *“This study shows that, during stop-signal task performance, children age 9–12 (n = 38) with more “adult-like” activity patterns show better inhibitory control performance. In addition, subthalamic nucleus activity and functional connectivity is related to inhibitory control in a developmental sample, but not in two adult samples (ns = 18 and 24). Although this study tackles an extremely interesting question with a clear, logical and multi-faceted approach, my enthusiasm is tempered by the relatively small developmental sample, which makes it difficult to disentangle developmental effects from individual differences. Additional replication data to support the claims that the “neural maturation index” reflects brain maturity per se and that the STN is involved in inhibitory control in development would significantly strengthen the manuscript. “*

Response: We are glad that the reviewer found that our analytical approach is clear and logical. Below we address changes made in response to the reviewer’s suggestions, noting that the focus of our study was to understand functional neural maturity during late childhood, a critical phase for development of cognitive control.

3.2 *“The small developmental sample and age range make it difficult to dissociate developmental effects from individual differences. Complementary approaches for disentangling these possibilities including testing a larger developmental sample and age range to show developmental change in inhibitory control (using, for example, Adolescent Brain Cognitive Development, IMAGEN, or PING data with a different measure of inhibitory control), and testing whether adults closer to the adult average template also show better SSRTs (this finding would suggest individual differences rather than developmental change is driving the result). Including data with other measures of inhibitory control could be particularly informative as previous work suggests that age only accounts for about 11% of the variance in SSRT (Bedard et al. 2002, Developmental Neuropsychology).”*

Response: We thank the reviewer for this comment and agree that it is important to study developmental change in inhibitory control from childhood to adolescence. Here we focused on a narrow but critical phase, for the development of inhibitory control, late childhood, based on well replicated behavioral findings (**Figure 1A, 1B**)^{1,2}. We implemented a novel multi-level analysis to understand functional maturity of neural circuitry underlying inhibitory control during this critical development period.

Our sample size (N=38) is equal to or larger than the majority of task fMRI studies of inhibitory control in children within a narrow age range (see **SI Table S1**) and in adults (see **SI Table S2**). Table S2 summarizes sample sizes of adult neuroimaging studies included in a previous meta-analysis, including 36 SST and 37 GNGT studies³. In fact, the sample size in the current study is larger than 66 out of the 70 adult inhibitory control studies. We agree that studying neurodevelopmental processes in a wider age range requires more participants; our study is unique in its focus on a narrow age range. In contrast, a recent study of cognitive control development focused on adolescents from 13 to 20 years old, splitting subjects into 4 age groups (13-14, 15-16, 17-18, 19-20 years old) with ~ 20 subjects per group, to examine behavioral performance by age⁴. Thus, our sample size of 38 compares well with previous studies in children. Importantly, we took a different angle here and focused on a larger sample in a narrow age range to address a specific developmental question: how functionally mature are the brains of 9-12 year olds and does more adult-like brain activation predict behavior.

While it would be ideal to test a larger developmental sample with a wider age range to show developmental changes in inhibitory control from late childhood to adulthood, such a dataset does not currently exist. The ABCD dataset at time point 1 only focuses on a very narrow age range (9-10 years old), IMAGEN dataset does not include children below 14 years old, and PING dataset does not have a Stop-Signal task. Furthermore, our Child cohort was well-suited to the neurodevelopmental questions posed in our study. We agree that additional studies needed to further understand the longitudinal developmental trajectory of functional circuitry underlying inhibitory control which we plan do with the ABCD data building on our current findings after release of ABCD data at time points 3 and 5 (ABCD time points 2 and 4 will not include neuroimaging data).

To test whether adults closer to the adult average template also show better SSRTs, we conducted additional analyses as suggested by the reviewer. However, we did not find a significant effect in either Adult cohort (OpenfMRI_Adult1: $p=0.17$; OpenfMRI_Adult2: $p=0.20$), highlighting a developmental specificity of the NMI. We have now included this result in the revised manuscript.

To further address the reviewer’s concern regarding other potential confounds in individual differences in developmental population, we included additional regressors that measure children’s cognitive development in the regression analyses. The additional regressors include Sequential Processing, Simultaneous Processing, Learning Ability, Planning Ability, and Mental Processing Index from Kaufman Assessment Battery for Children (KABC) (Supplementary Methods). After adding these additional regressors, we replicated our findings: (i) the NMI remains the most robust predictor (**SI Table S18**), (ii) STN activation is the most robust predictor (**SI Table S19**), (iii) the effective connectivity between rAI and rSTN is the most robust predictor for children’s SSRT (**SI Table S20**).

Table S18. Multiple linear regression analysis revealed that neural maturity index (NMI) predicts children’s SSRT, even after controlling effects of age, gender and head motion, KABC sequential processing, simultaneous processing, learning ability, planning ability and mental processing index.

	beta	t	p
Reference: Adults_OpenfMRI1			
Neural Maturity Index	-131.7	-1.94	0.06
Age	-0.9	-0.04	0.97
Gender	-11.8	-0.51	0.62
Maximum Frame-wise Displacement	-2.0	-0.16	0.87
KABC Sequential Processing	-4.3	-0.77	0.45
KABC Simultaneous Processing	-5.6	-0.96	0.34
KABC Learning Ability	-5.9	-1.05	0.30
KABC Planning Ability	-4.3	-0.88	0.39
KABC Mental Processing Index	15.1	0.94	0.36
Reference: Adults_OpenfMRI2			
Neural Maturity Index	-209.7	-2.70	0.01 *
Age	-4.4	-0.22	0.83
Gender	-7.4	-0.33	0.74

Maximum Frame-wise Displacement	-4.1	-0.34	0.74
KABC Sequential Processing	-5.6	-1.07	0.29
KABC Simultaneous Processing	-7.2	-1.29	0.21
KABC Learning Ability	-7.7	-1.42	0.17
KABC Planning Ability	-5.7	-1.22	0.23
KABC Mental Processing Index	19.5	1.27	0.22

Table S19. Multiple linear regression analysis revealed that STN activation is the most robust predictor for children’s SSRT after controlling effects of age, gender and head motion, KABC sequential processing, simultaneous processing, learning ability, planning ability and mental processing index.

	beta	t	p
Activation in rSTN	-4.7	-1.96	0.06
Age	-7.8	-0.36	0.72
Gender	-9.2	-0.40	0.69
Maximum Frame-wise Displacement	1.0	0.08	0.94
KABC Sequential Processing	-3.5	-0.64	0.53
KABC Simultaneous Processing	-4.4	-0.76	0.45
KABC Learning Ability	-4.6	-0.84	0.41
KABC Planning Ability	-3.4	-0.71	0.49
KABC Mental Processing Index	12.0	0.76	0.45
Activation in ISTN	-5.3	-1.83	0.08
Age	-0.9	-0.04	0.97
Gender	-3.6	-0.16	0.88
Maximum Frame-wise Displacement	-0.2	-0.02	0.99
KABC Sequential Processing	-3.0	-0.56	0.58
KABC Simultaneous Processing	-4.4	-0.75	0.46
KABC Learning Ability	-4.3	-0.78	0.44
KABC Planning Ability	-3.2	-0.67	0.51
KABC Mental Processing Index	11.3	0.71	0.48

Table S20. Multiple linear regression analysis revealed that effective connectivity between rAI and rSTN predicted children’s SSRT after controlling effects of age, gender and head motion, KABC sequential processing, simultaneous processing, learning ability, planning ability and mental processing index.

	beta	t	p
effective connectivity between rAI and rSTN during stopping	-21.3	-2.70	0.01 *

Age	-10.6	-0.52	0.61
Gender	-0.5	-0.02	0.98
Maximum Frame-wise Displacement	3.9	0.33	0.75
KABC Sequential Processing	-2.2	-0.42	0.68
KABC Simultaneous Processing	-2.7	-0.49	0.63
KABC Learning Ability	-2.5	-0.49	0.63
KABC Planning Ability	-1.8	-0.39	0.70
KABC Mental Processing Index	6.2	0.42	0.68
			
effective connectivity between rAI and ISTN during stopping	-1.5	-0.12	0.90
Age	-5.7	-0.24	0.81
Gender	-9.7	-0.39	0.70
Maximum Frame-wise Displacement	1.1	0.08	0.94
KABC Sequential Processing	-2.3	-0.40	0.70
KABC Simultaneous Processing	-3.2	-0.52	0.61
KABC Learning Ability	-3.6	-0.61	0.55
KABC Planning Ability	-2.5	-0.47	0.64
KABC Mental Processing Index	8.5	0.51	0.62

3.3 “The “neural maturation index” is defined as the spatial correlation between each child’s activation map and the average adult map in a mask defined in the adult data. This approach leaves open the possibility that children with higher NMIs actually show less similar patterns at the whole-brain level. Do results hold for whole-brain rather than ROI-based correlations?”

Response: To clarify, the neural maturity index (NMI) was computed using voxel-wise brain activation maps derived from adults, not from ROIs.

3.4 “Does the NMI reflect state-like or trait-like effects? The name suggests that it reflects trait-like developmental effects, but it is possible that it may reflect state-like effects such that trials/runs with good performance show more “typical” adult-like neural pattern and less successful trials/run show less “typical” patterns. Would this result change the authors’ interpretation of the results?”

Response: This is an interesting point, and we thank the reviewer for raising this. In the current study, it is difficult to conclude whether the NMI reflects a state-like versus a trait-like effect. NMI is a measure based on task fMRI data and likely depends on a subjects’ performance and their level of engagement in the task. Therefore, the NMI computed from some task runs in which subjects have low motivation and poor attention may indeed be different from the NMI computed from other task runs in which subjects have high motivation and good attention. Whether brain-behavior relationship depends on engagement of certain brain/cognitive states is a very interesting question. Our recent study has looked into dynamic brain states during cognitive task and demonstrated that individuals who engage more task-optimal brain states have better

behavioral performance⁵. Further research modulating children's motivation and attention across multiple runs of fMRI is needed to address the reviewer's question,

3.5 "In the developmental dataset, the motion exclusion thresholds of >5 mm displacement and >.5 mm average frame-to-frame displacement (is that what's mean by "scan-to-scan movement"?) are high compared to previous studies, which typically exclude runs with >2-3 mm maximum displacement and/or >.15-.3 mm mean frame-to-frame displacement. Are connectivity results consistent in individuals with low head motion, and consistent when controlling for mean (rather than maximum) framewise displacement?"

Response: We thank the reviewer for allowing us to clarify issues related to head motion. Indeed, this is a critical concern in all pediatric fMRI studies. While the exclusion thresholds of >0.5mm frame-to-frame displacement and > 5mm in overall displacement are not as strict as typical adult fMRI studies, this threshold is common in developmental studies. For example, a recent study published in *Nature Communications* used a similar head motion exclusion criterion in task fMRI: "Runs with a single-relative movement exceeding 5mm were excluded"⁴. Importantly, we have further clarified that frame-to-frame displacement was < 0.5mm (Supplementary Material), a benchmark now increasingly used in resting-state fMRI studies⁶. Task-based fMRI studies are less impacted.

To further address the reviewer's concern on head motion, we conducted additional analysis to censor scan volumes with excessive head motion using a similar procedure commonly implemented in previous developmental neuroimaging studies^{4,7}. Specifically, volumes with head motion exceeding 0.5 voxels or spikes in global signal exceeding 5% were interpolated using adjacent scans. Then, we conducted the same analyses to examine brain-behavior relation. We replicated all findings reported, summarize results below and report them in the Supplementary Results of the revised manuscript.

Neural maturity indices in children related to individual inhibitory control abilities

We found a significant negative correlation between the NMI and SSRT using a reference map derived from the OpenfMRI_Adult1 reference map ($r=-0.36$, $p=0.03$, *Cohen's d*=0.77) (**SI Figure S3**). We replicated this finding using the OpenfMRI_Adult2 reference map ($r=-0.41$, $p<0.05$, *Cohen's d*=0.89) (**SI Figure S3**). To further examine whether this relationship was driven by other potential confounds, we conducted multiple linear regression with SSRT as the dependent variable and NMI, age, gender, and maximum head motion displacement as independent variables. We found that the NMI is the most robust predictor (OpenfMRI_Adult1: $p<0.05$; OpenfMRI_Adult2: $p<0.05$) (**SI Table S14**).

Figure S3. NMI in children was negatively correlated with SSRT (with volume repair). This relationship was replicated using reference maps from the two adult cohorts. Each of the data points represents one child.

Table S14. Multiple linear regression analysis revealed that neural maturity index (NMI) predicts children’s SSRT, even after controlling effects of age, gender and head motion. Replication after volume repair analysis.

	beta	t	p
Reference: Adults_OpenfMRI1			
Neural Maturity Index	-122.1	-2.16	0.04*
Age	-11.8	-0.68	0.5
Gender	-4.84	-0.25	0.8
Maximum Frame-wise Displacement	-1.69	-0.16	0.88
Reference: Adults_OpenfMRI2			
Neural Maturity Index	-131.2	-2.52	0.02*
Age	-12.8	-0.75	0.46
Gender	-1.5	-0.08	0.94
Maximum Frame-wise Displacement	-3.1	-0.29	0.77

Children’s STN activation related to inhibitory action control ability

We found a negative correlation between STN activation in stopping and SSRT in the Stanford_Child dataset (ISTN: $r=-0.38$, $p=0.02$, *Cohen’s d*=0.82; rSTN: $r=-0.36$, $p=0.03$, *Cohen’s d*=0.77) (**Figure SI S4**). To further examine whether this relationship is driven by potential confounds, we conducted multiple linear regression with SSRT as the dependent variable and STN activation, age, gender and maximum head motion displacement as independent variables. We found that both rSTN and ISTN activation was the significant predictor ($ps<0.05$) after controlling effects of age, gender, and head motion (**SI Table S15**).

Figure S4. Activation in the STN during stopping was negatively correlated with SSRT in children (with volume repair). Each of the data points represents one child.

Table S15. Multiple linear regression analysis revealed that activation levels in the STN predicted children’s SSRT, even after controlling effects of age, gender and head motion. Replication after volume repair analysis.

	beta	t	p
Activation in rSTN	-5	-2.3	.03*
Age	-12.9	-0.75	0.46
Gender	-5.6	-0.3	0.77
Max Frame-wise Displacement	-0.9	-0.09	0.93
Activation in ISTN	-5.9	-2.24	.03*
Age	-6.4	-0.37	0.72
Gender	-3.3	-0.18	0.86
Max Frame-wise Displacement	-1.6	-0.15	0.88

Cortical-STN connectivity related to inhibitory control ability

We found that stop signal-modulated connectivity between the rAI and rSTN, but not the ISTN, during stopping was significantly correlated with SSRT ($r=-0.45$, $p=0.004$, *Cohen’s d*=1.0) in children (**SI Figure S5**), such that increased rAI-rSTN connectivity was associated with faster SSRTs. Connectivity between other prefrontal nodes and STN were not significantly correlated with SSRT ($p>0.05$). Multiple linear regression analyses, that included SSRT as the dependent variable and STN activation, age, gender, and maximum head motion displacement as independent variables, confirmed that task-modulated connectivity between the rAI and rSTN was the best predictor for SSRT after controlling effect of age, gender, and head motion ($p=0.007$) (**SI Table S16**).

Figure S5 Effective connectivity between the rAI (seed) and rSTN (target) was negatively correlated with SSRT in children. No such relation was observed in the left hemisphere (with volume repair). Each of the data points represents one child.

Table S16. Multiple linear regression analysis revealed that effective connectivity between rAI and rSTN predicted children’s SSRT after controlling effects of age, gender and head motion. Replication after volume repair analysis.

	beta	t	p
effective connectivity between rAI and rSTN during stopping	-18.81	-2.9	.007**
Age	-6.9	-0.4	0.68
Gender	2.4	0.13	0.9
Maximum Frame-wise Displacement	3.1	0.29	0.77
effective connectivity between rAI and ISTN during stopping	-8.6	-0.95	0.35
Age	-10.5	-0.57	0.57
Gender	-4.5	-0.23	0.82
Maximum Frame-wise Displacement	3.3	0.27	0.79

Including mean framewise displacement, rather than max framewise displacement, does not have a significant impact in the connectivity regression analyses. **SI Table S13** shows regression analysis results using mean framewise displacement.

Table S13. Multiple linear regression analysis revealed that effective connectivity between rAI and rSTN predicted children’s SSRT after controlling effects of age, gender and mean framewise head motion.

	beta	t	p
effective connectivity between rAI and rSTN during stopping	-20.3	-2.95	.006**
Age	-10.7	-0.65	0.52
Gender	2.2	0/13	0.9
Mean Frame-wise Displacement	86.8	0.61	0.55

effective connectivity between rAI and ISTN during stopping	-3.9	-0.4	0.69
Age	-11.8	-0.63	0.53
Gender	-4.5	-0.22	0.82
Mean Frame-wise Displacement	86.76	0.53	0.6

3.6 *“Were decisions about which contrasts to analyze made before analyses were performed? The motivation to report results from the successful stop vs. successful go contrast for activation and classification analyses, but the more-common successful vs. unsuccessful stop contrast for functional connectivity analysis, is unclear.”*

Response: We apologize that we did not make it clear in the manuscript. Both SuccStop-Go and SuccStop-UnsuccStop contrasts are widely used to study inhibitory control in functional neuroimaging studies. However, they target different cognitive/motor processes involved in stopping. We used the contrast of SuccStop-Go to examine brain activation similarity between children and adult and develop neural maturity index because it is widely used to examine neural substrates underlying the full process of stimulus-triggered stopping, including signal detection, triggering the stop process, and action cancellation. Based on the Race model ⁸, whether a prepotent response can be cancelled successfully depends on which process finishes at first. Therefore, SuccStop and UnsuccStop share some early stopping-related processes, such as signal detection and triggering stop process, which allows us to focus on differences in later stages of the stop process, such as action cancellation. The focus of our connectivity analyses was to test the hypothesis that the hyperdirect pathway linking the STN to the cortex is involved in this later stage of the stop process. Therefore, we specifically focused on whether task-evoked interaction between cortical regions and STN in SuccStop versus UnsuccStop is associated with individuals' stopping abilities. We have now clarified the choice of these contrasts in the revised manuscript. (see pages 15 & 17)

3.7 *“The STN results, while interesting, are relatively weak. Results in the two adult datasets do not replicate previous findings or the current developmental finding that SSRTs are inversely correlated with activation in STN activation during inhibitory control. STN classification and connectivity results are also not reported for adult samples. Replication in a larger developmental sample would increase confidence that (a) the STN was appropriately localized in children and (b) classification and individual differences results are robust.”*

Response: We thank the reviewer for this comment and feedback. Our finding of no correlation between STN activation and SSRT in adults was not surprising. Indeed, the majority of SST studies in adults have not reported a correlation between SSRT and STN (>34, based on meta-analysis study conducted in 2014 ³) with the exception of two ^{9,10}. These two studies had very small sample sizes for the correlation analysis (N=5 in ¹⁰, and N=15 in ⁹). Moreover, two recent studies found either no relation ¹¹ or borderline significant effects ¹². In our study, we report findings from two independent cohorts of 18 and 24 participants respectively.

To address the reviewer's comment about classification in Adult datasets, we repeated the same analyses that we conducted in the Stanford_Child cohort in the two Adult cohorts. We found significant cross-validation accuracy in OpenfMRI_Adult2 (rSTN: ACC=62.5%, $p=0.04$; ISTN: ACC=70.9%, $p=0.004$) but not in the OpenfMRI_Adult1 (rSTN: ACC=47.2%, $p=0.45$; ISTN: ACC=58.3%, $p=0.16$). These results have now included in the revised manuscript.

STN connectivity results for adult samples were reported in the original version of the manuscript: “*We repeated the same analyses in the two adult datasets but we did not find a significant correlation between task-modulated connectivity between the rAI and STN and SSRT ($p > 0.05$).*” To our knowledge, there are no previous studies that have reported significant correlation between task-evoked cortical-STN connectivity and stopping behavior in adults.

With regards to the lack of STN-related effects in adults, one possibility is that children may be more dependent on Stop-signal driven modulation of the hyperdirect cortical-STN pathway because of immature proactive and anticipatory control mechanisms. Adults, on the other hand, may better implement proactive control mechanisms, which is less dependent on the hyperdirect cortical-STN pathway. Further research is required to test this hypothesis.

While replication using larger sample sizes is important, the sample size in our study is comparable to other developmental neuroimaging studies (see **SI Table S1**), especially with a focus on a narrow age range to eliminate/reduce age-related confounds. While increasing the sample size can improve the replicability of results, an important consideration is that the replicability of task fMRI data is not solely contingent on a large sample size but also depends on the amount of individual-level sampling. Here, we used rigorous standards for inclusion of our child participants. Specifically, we required that each child participant completed two runs of SST, that consisted of 96 trials per run. This yielded a large number of trials that are comparable to adult fMRI studies of the SST^{10, 13}, these rigorous within-subject criteria, which have been shown to be a critical factor for producing replicable task fMRI findings¹⁴.

Lastly, increasing the sample size would not guarantee better spatial localization of the STN. The accurate localization of STN is better achieved using high spatial resolution MRI. Given its size and location, the STN is a difficult structure to localize with lower resolution 3T MRI. To this end, our STN localization was based on studies that used 7T MRI¹⁵, which has greater spatial resolution than the localization procedures used by previous studies^{9, 10}.

3.8 “The go/no-go task quantifies accuracy but not SSRT, whereas the stop-signal task quantifies SSRT but not accuracy (because task timing is individualized). A more accurate characterization of the tasks is that both have benefits and drawbacks for characterizing inhibitory control. “

Response: We agree that the GNGT quantifies accuracy and the stop-signal task quantifies SSRT. However, the SST can also provide a reliable metric of accuracy. For example, in many behavioral and some neuroimaging studies, the SST is implemented using one or multiple fixed stop-signal delays and stopping accuracy can be used to characterize inhibitory control, just like the go/no-go task¹⁶.

3.9 “Why does the meta-analytic finding suggest that, “in contrast to adults, extant functional neuroimaging studies of inhibitory control in children are less consistent”? Could this be driven by the inclusion of both SSTs and go/no-go tasks in the developmental meta-analysis? “

Response: We found weaker meta-analytic results in the developmental studies in comparison to studies in adults. It is unlikely due to the inclusion of both SST and GNGT in the children meta-analysis because both SST and GNGT are also included in the adult meta-analysis

studies. We believe that weak meta-analytic results in developmental studies may be due to small number of studies and small sample size in studies.

3.10 “It would be helpful for the authors to unpack what they mean by a “hyperdirect” pathway, which is a major focus of the text but never fully explained. In other words, what distinguishes a direct pathway from a hyperdirect pathway?”

Response: We are sorry that we did not make this clear in the manuscript. The term “hyperdirect” is used widely in the basal ganglia literature to describe a pathway linking the STN with the cortex, critically bypassing the striatum, which is a key component of the “direct” and “indirect” pathways of the basal ganglia ^{17, 18}. These pathways are involved in issuing and inhibiting motor responses.

Specifically, the *direct* pathway of basal ganglia is a cortico-striato-pallidal circuit underlying motor execution. Briefly, when a motor action is to be *executed*, cortical regions activate the striatum, which in turn inhibits the globus pallidus internal segment, thus releasing thalamic neurons from inhibition and exciting the cortex.

Conversely, the *indirect* pathway of the basal ganglia is important for response inhibition, perhaps more for a proactive and selective form of inhibitory control ¹⁹. Briefly, when a motor action is to be *inhibited*, cortical regions activate the striatum which inhibits the globus pallidus external segment, which then disinhibit the STN. The STN in turn increases firing rates of the globus pallidus internal segment, thus lower firing rates in thalamic neurons, which inhibits the cortex.

The *hyperdirect* pathway of basal ganglia is involved in rapid inhibitory control. Briefly, when a motor action is to be rapidly inhibited (as in the case with the SST when the stop cue suddenly appears), cortical regions excite the subthalamic nucleus, which in turn excites the globus pallidus internal segment, which inhibits thalamus. Thus, the hyperdirect pathway is a fast route for the cortex to quickly activate the STN for stopping actions bypassing the striatum, conferring a significant advantage for rapid stopping. We have clarified this description in our revised manuscript. Figure 1c further describes this pathway. (also see pages 3 & 12)

3.11” Correlational analyses should be clearly distinguished from predictive analyses. The term “predict” should be reserved for cross-validated models. “

Response: Thanks for the suggestion. For all correlational analyses, we have changed “predicts” to “related to” in the revised manuscript.

3.12 “In Figure 4D, it appears that significance is determined by parametrically converting r values to p values. However, data points (voxels) are non-independent, so the degrees of freedom are likely overestimated, compromising the validity of this conversion.”

Response: To clarify, we believe that this comment is about Figure 3D as there is no Figure 4D. To address the reviewer’s concern, we now report p-values based on a permutation test. Specifically, in each permutation, we randomly shuffled voxels in the adult brain activation maps. Then we computed correlation between the shuffled adult activation map and children activation map. We repeated the permutation procedure 100 times with different random seed each time and obtained a random distribution of spatial correlation between two brain activation

maps from which the p value is computed. We found that all of the correlations between children and adult correlation maps were significant ($p_s=0.01$, permutation test). We have accordingly updated the methods and relevant result sections in the revised manuscript.

3.13” *It would be helpful to keep the bar graph axes consistent in Figure 4C.*”

Response: We have made the change to Figure 4C as suggested by the reviewer.

3.14 “*In table S7 it would be helpful to include the number of voxels per ROI in the table*”

Response: We have included the number of voxels per ROI in the table S9 (in revised SI).

3.15 “*Rather than making the code available upon request it would be helpful to share the code in a publicly available repository (e.g., github) along with links to the publicly available datasets analyzed.*”

Response: We thank the reviewer for this suggestion. We will share the code on our lab website and post a download link next to the publication on our website.

(<https://med.stanford.edu/scsnl/publications.html>). Our recent study ⁵ is an example.

Reviewer #4

4.1 *“The goal of this study was to examine the neural mechanisms of inhibitory control using multiple approaches (meta analysis, open-source fMRI datasets, primary analysis). The use of multi-level approaches is a strength of the study. They found that by and large children recruit the same brain regions as adults during stopping. They also calculated a “neural maturity index”. The methods are sound, the driving questions are rooted in a strong theoretical foundation, and the conclusions are appropriate. However, my enthusiasm for this manuscript was dampened by the lack of clarity regarding the impact on the field. It is unclear how this study provides “new insights” about immature inhibitory control in the developing brain. Identification of the STN as important for inhibitory control in children is interesting but not transformative.”*

Response: We are pleased that the reviewer found our methods to be strong and conclusions to be appropriate. Below we provide responses to the reviewer’s comments and have revised the manuscript accordingly.

4.2 *“The meta-analysis is important but it is unclear how the inclusion criteria of 6-13 years of age was chosen. This broad age range includes adolescents and individuals undergoing puberty, both of which confound understanding of inhibitory control in children.”*

Response: We understand and echo the reviewer’s comment that a broad age range that includes children and adolescents can confound understanding of inhibitory control in children. However, there is a dearth of inhibitory control studies that exclusively target late childhood (9-12 years). Given this major restriction, we included studies where majority of participants were from the late childhood period. Moreover, we excluded all the developmental neuroimaging studies where majority of the participants were adolescents.

4.3 *“The primary data analysis of the Child SST contains a fairly small sample size at $n=38$. What were the effect sizes? Why were only 38 of the 78 participants who were recruited for the larger study included in this analysis?”*

Response: We thank the reviewers for this comment. Our sample size ($N=38$) is equal to or larger than the majority of task fMRI studies of inhibitory control in children within a narrow age range (see **SI Table S1**), the majority of studies in children and adolescents across a wider age range^{20, 21, 22, 23}, and also many adult studies (see **SI Table S2**). Table S2 summarizes sample sizes of adult neuroimaging studies included in a previous meta-analysis, including 36 SST and 37 GNGT studies³. In fact, the sample size in the current study is larger than 66 out of the 70 adult inhibitory control studies. A recent study of cognitive control development published in *Nature Communications* recruited 88 subjects 13 to 20 years old split into 4 age groups (13-14, 15-16, 17-18, 19-20 years) of ~ 22 subjects per group, to examine behavioral performance by age⁴. Moreover, 16 subjects were excluded in the fMRI analysis, leaving 72 samples across this large age-range.

In terms of effect size, these are reported in the manuscript. For example, for the correlation between rAI-rSTN gPPI weight and SSRT ($r=0.46$) and the Cohen’s d is 1.04. The estimated sample size required to achieve power of 0.8 at a significance level of 0.05 is 34.

We recruited a total of 78 children in the study. Two children did not complete the scan. Thirty-eight children were excluded in the final analysis because of excessive head motion (Subjects whose mean scan-to-scan movement were greater than 0.5mm⁶ in either run and/or maximum displacement exceeded 5mm²⁴ in either run and/or were excluded). Participants with poor behavioral performance, less than 80% accuracy on Go trials, or with greater than 80% or less than 20% accuracy on the Stop trials, or with longer RT in unsuccessful stop trials than go trials, in either fMRI run were also excluded. We have reported these details in the manuscript (Supplemental Information).

4.4 “A motion threshold of 5mm is quite high--with the norm being between 2-3mm in the field. How much data was lost to motion? and how many participants had greater than 3mm of motion displacement?”

Response: We thank the reviewer for allowing us to clarify issues related to head motion. Indeed, this is a critical concern in all pediatric fMRI studies. Importantly, we have further clarified that frame-to-frame displacement was < 0.5mm, a benchmark now increasingly used in resting-state fMRI studies in children and adults⁶. While the exclusion thresholds of >5mm displacement is not as strict as adult fMRI studies, this threshold is commonly used in developmental studies. For example, a recent study published in *Nature Communications* used a similar head motion exclusion criterion in task fMRI: “Runs with a single-relative movement exceeding 5mm were excluded”⁴. Task-based fMRI studies are less impacted.

Thirty-three out of the total 78 children were excluded in the analysis because of excessive head motion (Subjects mean scan-to-scan movement were greater than 0.5mm⁶ and/or whose maximum displacement exceeded 5 mm²⁴ in either run were excluded).

Twelve out of the remaining 38 children had greater than 3mm of motion displacement.

To further address the reviewer’s concern on head motion effect, we conducted additional analysis to censor scan volumes with excessive head motion using a similar procedure commonly implemented in previous developmental neuroimaging studies^{4,7}. Specifically, volumes with head motion exceeding 0.5 voxels or spikes in global signal exceeding 5% were interpolated using adjacent scans. Then, we conducted the same analyses to examine brain-behavior relation. We summarize these findings below and report them in the Supplementary Results of the revised manuscript.

Neural maturity indices in children related to individual inhibitory control abilities

We found a significant negative correlation between the NMI and SSRT using a reference map derived from the OpenfMRI_Adult1 reference map ($r=-0.36$, $p=0.03$, *Cohen’s d*=0.77) (**SI Figure S3**). We replicated this finding using the OpenfMRI_Adult2 reference map ($r=-0.41$, $p<0.05$, *Cohen’s d*=0.89) (**SI Figure S3**). To further examine whether this relationship was driven by other potential confounds, we conducted multiple linear regression with SSRT as the dependent variable and NMI, age, gender, and maximum head motion displacement as independent variables. We found that the NMI is the most robust predictor (OpenfMRI_Adult1: $p<0.05$; OpenfMRI_Adult2: $p<0.05$) (**SI Table S14**).

Figure S3. NMI in children was negatively correlated with SSRT (with volume repair). This relationship was replicated using reference maps from the two adult cohorts. Each of the data points represents one child.

Table S14. Multiple linear regression analysis revealed that neural maturity index (NMI) predicts children’s SSRT, even after controlling effects of age, gender and head motion. Replication after volume repair analysis.

	beta	t	p
Reference: Adults_OpenfMRI1			
Neural Maturity Index	-122.1	-2.16	0.04*
Age	-11.8	-0.68	0.5
Gender	-4.84	-0.25	0.8
Maximum Frame-wise Displacement	-1.69	-0.16	0.88
Reference: Adults_OpenfMRI2			
Neural Maturity Index	-131.2	-2.52	0.02*
Age	-12.8	-0.75	0.46
Gender	-1.5	-0.08	0.94
Maximum Frame-wise Displacement	-3.1	-0.29	0.77

Children’s STN activation related to inhibitory action control ability

We found a negative correlation between STN activation in stopping and SSRT in the Stanford_Child dataset (ISTN: $r=-0.38$, $p=0.02$, *Cohen’s d*=0.82; rSTN: $r=-0.36$, $p=0.03$, *Cohen’s d*=0.77) (**Figure SI S4**). To further examine whether this relationship is driven by potential confounds, we conducted multiple linear regression with SSRT as the dependent variable and STN activation, age, gender and maximum head motion displacement as independent variables. We found that both rSTN and ISTN activation was the significant predictor ($ps<0.05$) after controlling effects of age, gender, and head motion (**SI Table S15**).

Figure S4. Activation in the STN during stopping was negatively correlated with SSRT in children (with volume repair). Each of the data points represents one child.

Table S15. Multiple linear regression analysis revealed that activation levels in the STN predicted children’s SSRT, even after controlling effects of age, gender and head motion. Replication after volume repair analysis.

	beta	t	p
Activation in rSTN	-5	-2.3	.03*
Age	-12.9	-0.75	0.46
Gender	-5.6	-0.3	0.77
Max Frame-wise Displacement	-0.9	-0.09	0.93
Activation in ISTN	-5.9	-2.24	.03*
Age	-6.4	-0.37	0.72
Gender	-3.3	-0.18	0.86
Max Frame-wise Displacement	-1.6	-0.15	0.88

Cortical-STN connectivity related to inhibitory control ability

We found that stop signal-modulated connectivity between the rAI and rSTN, but not the ISTN, during stopping was significantly correlated with SSRT ($r=-0.45$, $p=0.004$, *Cohen’s d*=1.0) in children (**SI Figure S5**), such that increased rAI-rSTN connectivity was associated with faster SSRTs. Connectivity between other prefrontal nodes and STN were not significantly correlated with SSRT ($p>0.05$). Multiple linear regression analyses, that included SSRT as the dependent variable and STN activation, age, gender, and maximum head motion displacement as independent variables, confirmed that task-modulated connectivity between the rAI and rSTN was the best predictor for SSRT after controlling effect of age, gender, and head motion ($p=0.007$) (**SI Table S16**).

Figure S5 Effective connectivity between the rAI (seed) and rSTN (target) was negatively correlated with SSRT in children. No such relation was observed in the left hemisphere (with volume repair). Each of the data points represents one child.

Table S16. Multiple linear regression analysis revealed that effective connectivity between rAI and rSTN predicted children’s SSRT after controlling effects of age, gender and head motion. Replication after volume repair analysis.

	beta	t	p
effective connectivity between rAI and rSTN during stopping	-18.81	-2.9	.007**
Age	-6.9	-0.4	0.68
Gender	2.4	0.13	0.9
Maximum Frame-wise Displacement	3.1	0.29	0.77
effective connectivity between rAI and ISTN during stopping	-8.6	-0.95	0.35
Age	-10.5	-0.57	0.57
Gender	-4.5	-0.23	0.82
Maximum Frame-wise Displacement	3.3	0.27	0.79

4.5 “It is surprising that the authors only included two studies of the SST in the analysis, as several other groups have published fMRI studies of SST in children. Admittedly, many of these other manuscripts included other populations of children as well but the healthy comparison groups could have been included here.”

Response: We believe that this comment relates to the meta-analysis of developmental studies in inhibitory control. We conducted meta-analysis of developmental neuroimaging studies in the summer of 2018. We attempted a thorough search on PubMed with focus on the stop-signal task (SST) and go/no-go task (GNG) in children (6-13 years old). While we found many developmental studies as a result of this search, many did not pass the selection criterion to be included in the final meta-analysis. The selection criterion was reported in the manuscript: “Studies included in the meta-analysis were required to meet the following criteria: (1) study included children between 6 and 13 years of age; (2) study reported activation peaks in typically developing children, distinct from adult or clinical groups if any; (3) study used the SST or the GNGT; (4) study included activation contrast analysis that directly probes inhibitory control; (5)

study reported activation peaks using whole-brain analysis; and (6) the activations were reported in either Montreal Neurological Institute (MNI) or Talairach space.” (see page 14)

4.6 “The adult reference group data for the NMI calculation were derived from the two OpenfMRI adult datasets, which themselves contain a broad age range of participants, spanning 18-39 years and 18-33 years, respectively. This is problematic because of the significant neural changes occurring during the ages of 18 to roughly 25/26 years of age, which makes use of these datasets as ‘adult reference groups’ concerning”.

Response: We thank the reviewer for this comment. To our knowledge, the majority of neuroimaging studies in adults employ similar age ranges, unless there is explicit reason to focus on a narrower age range in adults. A previous lifespan study showed that SSRT in this age range (18-40 years old) is fairly stable², suggesting similar brain mechanisms for adults aged from 18 to 40 years old. Nevertheless, we acknowledge that these independent adult samples have a wider age range than the Stanford_Child samples we collected, but do not believe it impacts our findings.

To address this concern, we conducted additional analyses in which the age range in adults was restricted to 18 to 26 years old. In the OpenfMRI_Adult1 dataset, only 11 adults are under 27 years old, which has low power in the group level analysis and no voxel survived at corrected threshold ($p < 0.01$, FDR corrected). In the OpenfMRI_Adult2 dataset, there are 22 adults under 27 years old, which allows us to test whether our findings depend on the age range of adults .

We computed children’s neural maturity index (NMI) using a reference map derived from the subset (18-26 years old) of OpenfMRI_Adult2 reference map. We found a significant negative correlation between the NMI and SSRT ($r = -0.43$, $p = 0.007$, *Cohen’s d* = 0.95) (**SI Figure S6**). To further examine whether this relationship was driven by other potential confounds, we conducted multiple linear regression with SSRT as the dependent variable and NMI, age, gender, and maximum head motion displacement as independent variables. We found that the NMI is the most robust predictor ($p < 0.01$) (**SI Table S17**).

Figure S6. NMI in children was negatively correlated with SSRT (with volume repair). The reference map was derived from the subset (18-26 years old) of the OpenfMRI_Adult2 cohorts. Each of the data points represents one child.

Table S17. Multiple linear regression analysis revealed that neural maturity index (NMI) predicts children’s SSRT, even after controlling effects of age, gender and head motion.

The reference map was derived from the subset (18-26 years old) of the OpenfMRI_Adult2 cohorts.

	beta	t	p
Reference: Adults_OpenfMRI2 (18-26 years old)			
	-		
Neural Maturity Index	159.8	-2.74	0.009
Age	-13.1	-0.78	0.44
Gender	-2.2	-0.12	0.91
Maximum Frame-wise Displacement	-5.7	-0.54	0.59

Reference

1. Schachar R, Logan GD. Impulsivity and Inhibitory Control in Normal Development and Childhood Psychopathology. *Developmental psychology* **26**, 710-720 (1990).
2. Williams BR, Ponsesse JS, Schachar RJ, Logan GD, Tannock R. Development of inhibitory control across the life span. *Developmental psychology* **35**, 205-213 (1999).
3. Cai W, Ryali S, Chen T, Li CS, Menon V. Dissociable roles of right inferior frontal cortex and anterior insula in inhibitory control: evidence from intrinsic and task-related functional parcellation, connectivity, and response profile analyses across multiple datasets. *J Neurosci* **34**, 14652-14667 (2014).
4. Insel C, Kastman EK, Glenn CR, Somerville LH. Development of corticostriatal connectivity constrains goal-directed behavior during adolescence. *Nature communications* **8**, (2017).
5. Taghia J, *et al.* Uncovering hidden brain state dynamics that regulate performance and decision-making during cognition. *Nature communications* **9**, (2018).
6. Supekar K, *et al.* Deficits in mesolimbic reward pathway underlie social interaction impairments in children with autism. *Brain : a journal of neurology* **141**, 2795-2805 (2018).
7. Luculano T, *et al.* Cognitive tutoring induces widespread neuroplasticity and remediates brain function in children with mathematical learning disabilities. *Nature communications* **6**, 8453 (2015).
8. Logan GD, Cowan WB, Davis KA. On the ability to inhibit simple and choice reaction time responses: a model and a method. *Journal of experimental psychology Human perception and performance* **10**, 276-291 (1984).
9. Aron AR, Behrens TE, Smith S, Frank MJ, Poldrack RA. Triangulating a cognitive control network using diffusion-weighted magnetic resonance imaging (MRI) and functional MRI. *J Neurosci* **27**, 3743-3752 (2007).
10. Aron AR, Poldrack RA. Cortical and subcortical contributions to Stop signal response inhibition: role of the subthalamic nucleus. *J Neurosci* **26**, 2424-2433 (2006).
11. de Hollander G, Keuken MC, van der Zwaag W, Forstmann BU, Trampel R. Comparing Functional MRI Protocols for Small, Iron-Rich Basal Ganglia Nuclei Such as the Subthalamic Nucleus at 7 T and 3 T. *Human Brain Mapping* **38**, 3226-3248 (2017).
12. Jahfari S, Ridderinkhof KR, Collins AGE, Knapen T, Waldorp LJ, Frank MJ. Cross-Task Contributions of Frontobasal Ganglia Circuitry in Response Inhibition and Conflict-Induced Slowing. *Cereb Cortex*, (2018).
13. Xue G, Aron AR, Poldrack RA. Common neural substrates for inhibition of spoken and manual responses. *Cereb Cortex* **18**, 1923-1932 (2008).
14. Nee DE. fMRI replicability depends upon sufficient individual-level data. *bioRxiv*, (2018).

15. Forstmann BU, *et al.* Cortico-subthalamic white matter tract strength predicts interindividual efficacy in stopping a motor response. *Neuroimage* **60**, 370-375 (2012).
16. Leung HC, Cai W. Common and differential ventrolateral prefrontal activity during inhibition of hand and eye movements. *J Neurosci* **27**, 9893-9900 (2007).
17. Nambu A. A new dynamic model of the cortico-basal ganglia loop. *Prog Brain Res* **143**, 461-466 (2004).
18. Nambu A, Tokuno H, Takada M. Functional significance of the cortico-subthalamo-pallidal 'hyperdirect' pathway. *Neurosci Res* **43**, 111-117 (2002).
19. Aron AR. From reactive to proactive and selective control: developing a richer model for stopping inappropriate responses. *Biol Psychiatry* **69**, e55-68 (2011).
20. Bhaijiwala M, Chevrier A, Schachar R. Withholding and canceling a response in ADHD adolescents. *Brain Behav* **4**, 602-614 (2014).
21. Pliszka SR, Glahn DC, Semrud-Clikeman M, Franklin C, Perez R, Xiong JJ. Neuroimaging of inhibitory control areas in children with attention deficit hyperactivity disorder who were treatment naive or in long-term treatment. *Am J Psychiat* **163**, 1052-1060 (2006).
22. Rubia K, Halari R, Cubillo A, Mohammad AM, Scott S, Brammer M. Disorder-specific inferior prefrontal hypofunction in boys with pure attention-deficit/hyperactivity disorder compared to boys with pure conduct disorder during cognitive flexibility. *Hum Brain Mapp* **31**, 1823-1833 (2010).
23. Durston S, Mulder M, Casey BJ, Ziermans T, van Engeland H. Activation in ventral prefrontal cortex is sensitive to genetic vulnerability for attention-deficit hyperactivity disorder. *Biol Psychiat* **60**, 1062-1070 (2006).
24. Chang TT, Rosenberg-Lee M, Metcalfe AW, Chen T, Menon V. Development of common neural representations for distinct numerical problems. *Neuropsychologia* **75**, 481-495 (2015).

Reviewers' Comments:

Reviewer #1:

Remarks to the Author:

The authors have effectively addressed all of the issues raised in the initial review. I have no further concerns.

Reviewer #2:

Remarks to the Author:

The authors have responded very well to the points I raised. I feel that the ms is much improved and accessible, now embracing a broader range of issues raised by their data. A very good ms has been converted into a truly excellent one that will be much cited. I have no further points to make.

Reviewer #3:

Remarks to the Author:

I thank the authors for addressing my comments about motion and other confounds and adding nonparametric significance testing. However, several of my major concerns have not been addressed in the text.

(1) My primary concern, that the current analyses cannot disentangle developmental variability from individual variability, is a significant caveat to the current results that is not mentioned at all in the manuscript. It is important to be transparent about the fact that there are at least two possible explanations of the results: (1) that developmental change drives the effect that children with better performance show more "adult-like" brain activity patterns, such that children's performance improvements would be reflected in increasing NMIs across development; or (2) that any participant with a more "typical" brain activity pattern related to stopping would show a better SSRT, such that the current result is not related to development or maturity at all. Adding the neural maturation index analyses of the adult samples as well as a discussion of this issue to the manuscript could help address this concern.

(2) The authors cite previous work to argue that their sample size of 38 is large compared to previous literature. Although I appreciate the challenges associated with scanning developmental populations, I am not convinced that the sample size is acceptable because it is based on outdated norms set by funding and time constraints rather than power analyses of effects between brain and behavioral measures. The ABCD Study has now made SST fMRI data from over 11,000 children publicly available. The authors suggest that the small age range (9-10) of the children in this study is a disadvantage when in fact testing their analysis on many children of the same age would be enormously beneficial in understanding whether the "maturational index" reflects developmental change or individual differences. If the authors are unwilling to include this well-powered sample in their analyses it should at least be acknowledged as a future direction.

(3) It should be acknowledged in the manuscript that the SST does not provide a holistic measure of inhibitory control in development or adulthood. Rather, it indexes one aspect of response inhibition that does not necessarily relate to other measures of inhibitory control (e.g., King et al., 2014, Current Addiction reports; Eisenberg, Bissett, et al., 2018, bioRxiv).

(4) On line 105, the authors write that in adolescence "impulsive and risky behaviors become

increasingly deleterious." It would be helpful to add more nuance here, since risk taking not necessarily more deleterious in one developmental period than another. Rather, it is more common in certain contexts in adolescence and may actually be more beneficial during this period (e.g., exploration, establishing independence).

Reviewer #4:

Remarks to the Author:

The authors have done a great job addressing the majority of reviewer concerns but a few still remain. Namely, the remaining concerns center around age and development, questions that were raised by all reviewers.

for example, R4 questioned the adult reference group. In response to this concern, the authors noted that a previous lifespan study 'showed that SSRT in this age range (18-40 years old) is fairly stable' (Williams et al, 1999). However, that is not exactly correct. The Williams paper reports: 'Results indicated the speed of stopping becomes faster with increasing age throughout childhood, with limited evidence of slowing across adulthood. By contrast, strong evidence was obtained for age-related speeding of go-signal reaction time throughout childhood, followed by marked slowing throughout adulthood,' suggesting that there was change in that study throughout adulthood.

A second issue is about motion. They lost data from 38 children because of excessive head motion. This is a high number because it means they excluded 50% of the subjects recruited for the study even at a fairly generous threshold of maximum displacement that exceeded 5mm. It makes one wonder 1) was the task too boring for children to stay still?; 2) Were the research assistants acquiring the data not trained to work with developmental populations; or 3) Are the subjects included in the analysis not representative of typically developing children? The last question is the most concerning.

A third issue remains about contribution to the field. The authors did not address this comment: "Identification of the STN as important for inhibitory control in children is interesting but not transformative."

Reviewer #1

1.1 The authors have effectively addressed all of the issues raised in the initial review. I have no further concerns.

Response: We are glad that the reviewer found our study to be suitable for publication in Nature Communications.

Reviewer #2

2.1 The authors have responded very well to the points I raised. I feel that the ms is much improved and accessible, now embracing a broader range of issues raised by their data. A very good ms has been converted into a truly excellent one that will be much cited. I have no further points to make.

Response: We thank the reviewer for the compliment.

Reviewer #3

3.1 I thank the authors for addressing my comments about motion and other confounds and adding nonparametric significance testing. However, several of my major concerns have not been addressed in the text.

Response: We are glad that the reviewer found that our additional analyses from the last revision successfully addressed head motion and other confounds. Here we address the reviewer's remaining comments, including replication using the ABCD dataset.

3.2 My primary concern, that the current analyses cannot disentangle developmental variability from individual variability, is a significant caveat to the current results that is not mentioned at all in the manuscript. It is important to be transparent about the fact that there are at least two possible explanations of the results: (1) that developmental change drives the effect that children with better performance show more "adult-like" brain activity patterns, such that children's performance improvements would be reflected in increasing NMIs across development; or (2) that any participant with a more "typical" brain activity pattern related to stopping would show a better SSRT, such that the current result is not related to development or maturity at all. Adding the neural maturation index analyses of the adult samples as well as a discussion of this issue to the manuscript could help address this concern.

Response: This is an excellent point. As suggested by the reviewer, we conducted a similar analysis within the adult samples and found that adults did not show a relation between NMI and SSRT in either adult cohort (OpenfMRI_Adult1: $p=0.17$; OpenfMRI_Adult2: $p=0.20$). This result highlights the developmental specificity of the NMI. We have now included the findings in the manuscript (see Page 9).

We further note that more fully disentangling developmental and individual variability will require longitudinal data, which will be facilitated by future releases of ABCD data release in Years 3 and 5. Our findings provide a novel template and suggest new

research directions in this regard. These points have now been noted in the Discussion (see page 12).

“There are two possible explanations for our findings: (1) developmental change drives the effect that children with better performance show more “adult-like” brain activity patterns; or (2) that any participant with a more “typical” brain activity pattern related to stopping would show a better SSRT, such that our NMI results are not related to development or maturity. To address this question, we conducted a similar analysis within the adult samples and found that adults did not show a relation between NMI and SSRT. These results suggest that NMI results are related to development and maturation and supports the first hypothesis. Studies using NMI measures, as developed here, and longitudinal data are needed to further disentangle sources of individual variability over development. Our findings provide a novel template for new research directions in this regard.”

3.3 The authors cite previous work to argue that their sample size of 38 is large compared to previous literature. Although I appreciate the challenges associated with scanning developmental populations, I am not convinced that the sample size is acceptable because it is based on outdated norms set by funding and time constraints rather than power analyses of effects between brain and behavioral measures. The ABCD Study has now made SST fMRI data from over 11,000 children publicly available. The authors suggest that the small age range (9-10) of the children in this study is a disadvantage when in fact testing their analysis on many children of the same age would be enormously beneficial in understanding whether the “maturational index” reflects developmental change or individual differences. If the authors are unwilling to include this well-powered sample in their analyses it should at least be acknowledged as a future direction.

Response: To demonstrate the robustness of our findings, we analyzed SST fMRI data from the ABCD study¹, as the reviewer suggested. Due to limitations of computing and storage resources, we analyzed SST fMRI data from the first 500 typically developing children (TDC). Our analyses using fMRI and behavioral data from children ages 9-11 in the ABCD study replicated all major results from the Stanford_Child dataset, thereby demonstrating the robustness of our findings. We have summarized the main replication findings in the Results section (see pages 9-10), and included figures and tables in the Supplemental Information.

We downloaded minimal preprocessed data from the ABCD study (two SST sessions per subject), which we then normalized to Montreal Neurological Institute (MNI) 2mm template (91x109x91) and smoothed using 6mm Gaussian kernel. We created a task design matrix using the event and onset data provided by the ABCD study and ran the same GLM and gPPI analyses.

Because of missing data or event conditions, we could not complete GLM analysis on 17 participants. Among the remaining 483 participants, we excluded 240 participants because their head motion displacement was over 5mm (the criterion we used in our main analysis). We excluded individuals who had poor behavioral performance (less than 80% accuracy on Go trials, or with greater than 80% or less than 20% accuracy on Stop trials, or with longer RT in unsuccessful stop trials than go trials) and individuals who have outliers in key behavioral measure (i.e. SSRT) and brain measures (e.g. NMI,

STN activation and PPI weight). Outliers was defined by more than 3 standard deviations away from the mean. The final sample included 186 TDC (9-11 years old, 104F/82M).

First, we examined the relationship between NMI and SSRT in the ABCD dataset. We found a significant negative correlation between the NMI and SSRT ($r=-0.15$, $p<0.05$, *Cohen's d*=0.30) (**Figure S7**). To further examine whether this relationship was driven by other potential confounds, we conducted multiple linear regression with SSRT as the dependent variable and NMI, age, gender, and maximum head motion displacement as independent variables. We found that the NMI was the most robust predictor ($p=0.04$) (**Table S21**).

Figure S7. NMI in children was negatively correlated with SSRT (replication using the ABCD dataset). Each data point represents one child. Source data are provided as a Source Data file.

Table S21. Multiple linear regression analysis revealed that neural maturity index (NMI) predicts children's SSRT, after controlling for age, gender and head motion. Replication using the ABCD dataset.

	beta	t	p
	-		
Neural Maturity Index	130.36	-2.02	0.04*
Age	-0.98	-1.73	0.08
Gender	1.96	0.22	0.82
Maximum Frame-wise Displacement	4.59	1.16	0.25

Second, we examined the relationship between STN activation during stopping and SSRT in the ABCD dataset. We found a negative correlation between STN activation in stopping and SSRT (ISTN: $r=-0.22$, $p=0.005$, *Cohen's d*=0.45; rSTN: $r=-0.20$, $p=0.005$, *Cohen's d*=0.41) (**Figure S8**). To further examine whether this relationship is driven by potential confounds, we conducted multiple linear regression with SSRT as the dependent variable and STN activation, age, gender, and maximum head motion displacement as independent variables. We found that STN activation was the most robust predictor of SSRT (ISTN: $p=0.0002$; rSTN: $p=0.0006$) after controlling effects of age, gender, and head motion (**Table S22**).

Figure S8. STN activation during Stopping was negatively correlated with SSRT in children (replication using the ABCD dataset). Each data point represents one child. Source data are provided as a Source Data file.

Table S22. Multiple linear regression analysis revealed that activation levels in the STN predicted children’s SSRT, controlling for effects of age, gender and head motion. Replication using the ABCD dataset.

	beta	t	p
Activation in rSTN	-2.98	-3.46	0.0007***
Age	-1.36	-2.4	0.02*
Gender	-4	-0.47	0.64
Max Frame-wise Displacement	5.6	1.45	0.15
Activation in ISTN	-3.22	-3.79	0.0002***
Age	-1.43	-2.52	0.01*
Gender	-4.81	-0.56	0.58
Max Frame-wise Displacement	5.53	1.44	0.15

Third, we examined the relationship between effective connectivity between rAI and STN and SSRT in the ABCD dataset. We found that stop signal-modulated connectivity between the rAI and rSTN, but not the ISTN, during stopping was significantly correlated with SSRT ($r=-0.16$, $p<0.05$, *Cohen’s* $d=0.32$) in children (**Figure S9**), such that increased rAI-rSTN connectivity was associated with faster SSRTs. Multiple linear regression analyses, which included SSRT as the dependent variable and STN activation, age, gender, and maximum head motion displacement as independent variables, confirmed that task-modulated connectivity between the rAI and rSTN was the best predictor for SSRT ($p=0.03$) (**Table S23**).

Figure S9. Effective connectivity between the rAI (seed) and rSTN (target) was negatively correlated with SSRT in children. No such relation was observed in the left hemisphere (replication using the ABCD dataset). Each data point represents one child. Source data are provided as a Source Data file.

Table S23. Multiple linear regression analysis revealed that effective connectivity between rAI and rSTN predicted children’s SSRT after controlling for age, gender and head motion. Replication using the ABCD dataset.

	beta	t	p
Effective connectivity between rAI and rSTN during stopping	-9.46	-2.14	0.04*
Age	-0.93	-1.65	0.1
Gender	0.65	0.08	0.94
Maximum Frame-wise Displacement	5.02	1.28	0.2
Effective connectivity between rAI and ISTN during stopping	-5.32	-1.19	0.23
Age	-0.92	-1.61	0.11
Gender	-2.08	-0.24	0.81
Maximum Frame-wise Displacement	5.14	1.29	0.19

3.4 *It should be acknowledged in the manuscript that the SST does not provide a holistic measure of inhibitory control in development or adulthood. Rather, it indexes one aspect of response inhibition that does not necessarily relate to other measures of inhibitory control (e.g., King et al., 2014, Current Addiction reports; Eisenberg, Bissett, et al., 2018, bioRxiv).*

Response: We have acknowledged in the revised manuscript that SST is one of the many paradigms for studying inhibitory control and SSRT indexes one aspect of response inhibition (see page 11).

3.5 *On line 105, the authors write that in adolescence “impulsive and risky behaviors become increasingly deleterious.” It would be helpful to add more nuance here, since risk taking not necessarily more deleterious in one developmental period than another. Rather, it is more common in certain contexts in adolescence and may actually be more beneficial during this period (e.g., exploration, establishing independence).*

Response: We thank the reviewer for this comment. We have now changed “impulsive and risky” to “impulsive and maladaptive” in the revised manuscript (see page 3).

Reviewer #4

4.1 *The authors have done a great job addressing the majority of reviewer concerns but a few still remain. Namely, the remaining concerns center around age and development,*

questions that were raised by all reviewers.

Response: We are glad that the reviewer found that our last response letter answered the main questions raised by the reviewer. Here we address the remaining comments.

4.2 for example, R4 questioned the adult reference group. In response to this concern, the authors noted that a previous lifespan study 'showed that SSRT in this age range (18-40 years old) is fairly stable' (Williams et al, 1999). However, that is not exactly correct. The Williams paper reports: 'Results indicated the speed of stopping becomes faster with increasing age throughout childhood, with limited evidence of slowing across adulthood. By contrast, strong evidence was obtained for age-related speeding of go-signal reaction time throughout childhood, followed by marked slowing throughout adulthood,' suggesting that there was change in that study throughout adulthood.

Response: We thank the reviewer for this feedback. Our point was really quite the same: “the speed of stopping becomes faster with increasing age throughout childhood with limited evidence of slowing across adulthood”. A closer inspection of their data reveals that while SSRT changes from young adulthood (18-29 years old) to the elderly (60-81 years old), the point that is relevant for our study is the 18-44 age range of our Adult cohorts. As reported by Williams et al. (1999): SSRT=208.6±75.1ms in 18-29 year old adults and SSRT=209.7±63.1ms in 30-44 year old adults.

4.3 A second issue is about motion. They lost data from 38 children because of excessive head motion. This is a high number because it means they excluded 50% of the subjects recruited for the study even at a fairly generous threshold of maximum displacement that exceeded 5mm. It makes one wonder 1) was the task too boring for children to stay still?; 2) Were the research assistants acquiring the data not trained to work with developmental populations; or 3) Are the subjects included in the analysis not representative of typically developing children? The last question is the most concerning.

Response: We thank the reviewer for comments on the high ratio of exclusion due to excessive head motion. We do not think such levels of dropout are specific to our study². In the ABCD dataset used for replication in the revised manuscript, for example, we found that 240 out of 483 typically developing children had greater than 5mm maximal displacement. The dropping rate is 49.7%, as shown in the histogram below:

Whether the sample in a research study is representative of the targeted population is a challenge question that most development and clinical studies face. For example, can children with low head motion be representative of children with high head motion? Can clinical patients with low head motion be representative to patients with high head motion? Unfortunately, we do not know the answer because we cannot get reliable results from individual with excessive head motion.

4.4 A third issue remains about contribution to the field. The authors did not address this comment: "Identification of the STN as important for inhibitory control in children is interesting but not transformative."

Response: We are sorry that we did not directly address this comment directly in our last response letter. As noted in the manuscript, while the STN has been well studied in animal models, only a few functional neuroimaging studies in adults have examined STN function in neurotypical adults and there have been no studies in children. More generally, little is known about the functional maturation of cortical and basal ganglia systems, their interactions, and their relation to inhibitory control ability in children. We believe that identification of the STN as a key node and the cortical-STN as an important circuit for inhibitory control is crucial for understanding the development of inhibitory control in children. Importantly, it moves us away from a cortico-centric view of inhibitory control. Our findings provide a novel template for investigating cortico-basal ganglia dysfunction in ADHD. STN measures are also likely to contribute novel features to improve diagnosis and evaluation of treatment effect in ADHD. We have now noted these points in the revised manuscript.

Reference

1. Casey BJ, *et al.* The Adolescent Brain Cognitive Development (ABCD) study: Imaging acquisition across 21 sites. *Dev Cogn Neurosci* **32**, 43-54 (2018).
2. Yerys BE, *et al.* The fMRI success rate of children and adolescents: typical development, epilepsy, attention deficit/hyperactivity disorder, and autism spectrum disorders. *Hum Brain Mapp* **30**, 3426-3435 (2009).

Reviewers' Comments:

Reviewer #3:

Remarks to the Author:

I thank the authors for addressing my concerns. My only remaining comment is to acknowledge the ABCD data and create a NIMH Data Repositories Study as outlined in the ABCD Data Use Certification Terms and Conditions #6 and 9.

Reviewer #4:

Remarks to the Author:

The authors have done a thorough job addressing remaining reviewer concerns but my concern about a novel contribution remains. I defer to the editors

Reviewer #3:

I thank the authors for addressing my concerns. My only remaining comment is to acknowledge the ABCD data and create a NIMH Data Repositories Study as outlined in the ABCD Data Use Certification Terms and Conditions #6 and 9.

Response: We have acknowledged the ABCD data in the revised manuscript and will create a NIMH Data Repositories Study.

Reviewer #4:

The authors have done a thorough job addressing remaining reviewer concerns but my concern about a novel contribution remains. I defer to the editors

Response: Our study has multiple novel contributions to understanding development of inhibitory control function in children. First, while the late childhood is important for development of inhibitory control function, maturation of brain mechanism in this critical developmental stage and its association with children's inhibitory control function is unknown. Here, we use a novel neural maturity index to demonstrate that the functional maturation of cortical and basal ganglia systems is related to inhibitory control ability in children. Second, while the STN has been well studied in animal models, only a few functional neuroimaging studies in adults have examined STN function in neurotypical adults and there have been no studies in children. In this study, we demonstrate that the STN is a key node for inhibitory control mechanism in children. Third, we believe that the cortical-STN as an important circuit for inhibitory control is crucial for understanding the development of inhibitory control in children. Importantly, it moves us away from a cortico-centric view of inhibitory control. Our findings provide a novel template for investigating cortico-basal ganglia dysfunction in ADHD. STN measures are also likely to contribute novel features to improve diagnosis and evaluation of treatment effect in ADHD.